# BACK TO BASICS:
# EFFICIENT NETWORK COMPRESSION VIA IMP

## ABSTRACT

Network pruning is a widely used technique for effectively compressing Deep Neural Networks with little to no degradation in performance during inference. Iterative Magnitude Pruning (IMP) (Han et al., 2015) is one of the most established approaches for network pruning, consisting of several iterative training and pruning steps, where a significant amount of the network's performance is lost after pruning and then recovered in the subsequent retraining phase. While commonly used as a benchmark reference, it is often argued that a) its iterative nature makes it slow and non-competitive, b) it reaches suboptimal states, in particular because it does not incorporate sparsification into the training phase, and c) its global selection criterion fails to properly determine optimal layer-wise pruning rates. In light of recently proposed retraining techniques, we investigate these claims through rigorous and consistent experiments. We find IMP to be surprisingly efficient, achieving its full potential with significantly less than the extensive amount of retraining usually considered necessary (Renda et al., 2020). When paired with the compressed learning rate scheme suggested by Le & Hua (2021), we find that it can not only perform on par with more complex or heavily parameterized state-of-the-art approaches, but it does so without or with only little computational overhead even when using its original global magnitude selection criterion. This casts doubt on the commonly claimed advantages of imposing an implicit bias during training to avoid retraining.

## 1 INTRODUCTION

Modern Neural Network architectures are commonly highly over-parameterized (Zhang et al., 2016), containing millions or even billions of parameters, resulting in both high memory requirements as well as computationally intensive and long training and inference times. It has been shown however (LeCun et al., 1989; Hassibi & Stork, 1993; Han et al., 2015; Gale et al., 2019; Lin et al., 2020; Blalock et al., 2020) that modern architectures can be compressed dramatically by *pruning*, i.e., removing redundant structures such as individual weights, entire neurons or convolutional filters. The resulting *sparse* models require only a fraction of storage and floating-point operations (FLOPs) for inference, while experiencing little to no degradation in predictive power compared to the dense model. There is of course an inherent tradeoff between sparsity and model performance; a very heavily pruned model will normally be less performant than its dense (or moderately pruned) counterpart, though it has been observed that pruning might have a regularizing effect and be beneficial to the generalization capacities (Blalock et al., 2020; Hoefler et al., 2021).

One approach to pruning consists of removing part of a network's weights after a standard training process, seemingly losing most of its predictive performance, and then retraining to compensate for that pruning-induced loss. This can be done either once (*One Shot*), or the process of pruning and retraining can be repeated iteratively until the desired level of sparsity is reached. Although dating back to the early work of Janowsky (1989), this approach was most notably proposed by Han et al. (2015) in the form of ITERATIVE MAGNITUDE PRUNING (IMP). Because it is arguably one of the simplest pruning algorithms, IMP has been widely applied as a baseline comparison for other approaches (Carreira-Perpiñán & Idelbayev, 2018; Ding et al., 2019; Savarese et al., 2020; Siegel et al., 2020; Hoefler et al., 2021). As such, it is often subject to criticism, with the most commonly made claims arguing against the efficacy of IMP being the following:

1.) Inherent to the justification of many proposed alternatives is the claim, that *sparsification should be part of the training*. Methods of this type reach a sparse model at the end of training, ideally eliminating the need for further training. One desired benefit of doing so, is to improve the sparsity vs. performance tradeoff by reducing the impact of the actual 'hard' pruning, which results in a "failure to properly recover the pruned weights" (Liu et al., 2020). It is argued that IMP achieves sub-optimal states since learning the pruning set throughout training "helps find a better subset and hence prune more weights with no or little loss degradation" (Carreira-Perpiñán & Idelbayev, 2018). Another frequently claimed advantage is that incorporating the sparsification into the training cuts down on computational cost by not requiring additional retraining epochs. Ding et al. (2019) for example advertise that there is "no need for a time consuming re-training" and Hoefler et al. (2021) argue that "the sparsify-during-training schedule (...) is usually cheaper than the train-then-sparsify schedule".

2.) IMP determines a single numerical threshold for pruning and applies it globally to every parameter, potentially resulting in very different levels of sparsity among the layers of the network. This behavior is often considered to be sub-optimal and it is argued that *pruning should be layer-dependent* (Liu et al., 2020), so more complex saliency criteria have been proposed (Gale et al., 2019; Lee et al., 2020). Kusupati et al. (2020) for example claim that "uniform or heuristic non-uniform sparsity budgets (...) have sub-optimal layer-wise parameter allocation resulting in a) lower prediction accuracy or b) higher inference cost (FLOPs)".

3.) While only the iterative approach, that is repeatedly removing only a small fraction of the parameters followed by extensive retraining, is said to achieve results on the Pareto frontier (Renda et al., 2020), its iterative nature is also considered to be *computationally tedious, if not impractical*: "iterative pruning is computationally intensive, requiring training a network 15 or more times consecutively for multiple trials" (Frankle & Carbin, 2018), leading Liu et al. (2020) to trying to "avoid the expensive pruning and fine-tuning iterations".

Our interest lies in exploring these claimed disadvantages of IMP through rigorous and consistent computational experimentation with a focus on recent advancements concerning the retraining phase, see the results of Renda et al. (2020) and Le & Hua (2021). This comparative study is in fact intended to complement both of these works, which focused on improving the sparsity-vs.-performance tradeoff of IMP through improved learning rate schemes during training, by putting an additional spotlight on the total computational cost of IMP in a direct comparison with methods that are commonly assumed to outperform IMP in that aspect by avoiding retraining.

**Contributions.** We empirically find that, using an appropriate learning rate scheme, only few retraining epochs are needed in practice to achieve most of the sparsity vs. performance tradeoff of IMP. We also find that the global selection criterion not only finds sparsity distributions on par with but, somewhat surprisingly, often better than those of more sophisticated layer-dependent pruning criteria. Finally, we conclude that, using an appropriate learning rate scheme, IMP performs well even when compared to state-of-the-art approaches that incorporate sparsification into the training *without or with only little computational overhead*. That is, not only can IMP find some of the best performing architectures at any given sparsity level, but due to the compressed retraining time it does so without needing to leverage a longer running time even when compared to methods typically considered to be superior to IMP in that particular aspect.

**Outline.** Section 2 contains a complete overview over related works, including a brief summary of all pruning methods and approaches considered here. In Section 3 we provide the computational results and their interpretation by first addressing how IMP can develop its full potential within a restricted computational envelope in Subsection 3.1 and Subsection 3.2. We then use the resulting lessons in order to draw up a fair comparison to methods that incorporate pruning into their training in Subsection 3.3. We conclude with some discussion in Section 4.

## 2 OVERVIEW OF PRUNING METHODS AND METHODOLOGY

While the sparsification of Neural Networks includes a wide variety of approaches, we will focus on *Model Pruning*, i.e., the removal of redundant structures in a Neural Network. More specifically, our results will be limited to *unstructured* pruning, that is the removal of individual weights, as opposed

to its *structured* counterpart, where entire groups of elements, such as neurons or convolutional filters, are removed. We will also focus on approaches that start with a dense network and then either prune the network *during* training or *after* training as already discussed in the introduction. Following Bartoldson et al. (2020), we will also refer to methods of the former category as *pruning stable*, since the final pruning should result in a negligible decrease in performance, where methods of the latter category are referred to as *unstable*. For a full and detailed survey of Pruning algorithms we refer the reader to Hoefler et al. (2021).

Pruning unstable methods are exemplified by ITERATIVE MAGNITUDE PRUNING (IMP) (Han et al., 2015). In its original form, it first employs standard network training, adding a common $\ell_2$-regularization term on the objective, and then removes all weights from the network whose absolute values are below a certain threshold. The network at this point commonly loses some or even all of its learned predictive power, so it is then retrained for a fixed number of epochs. This prune-retrain cycle is usually repeated a number of times; the threshold at every pruning step is determined as the appropriate percentile such that, at the end of given number of iterations, a desired target sparsity is met.[1] In the following we will first discuss two particular details of IMP that have been the focus of recent research: the questions of (a) how to select the parameters to be pruned and (b) how to retrain. We will then conclude this section by briefly outlining the pruning stable methods we have selected for this comparison and establish how to fairly compare them to IMP.

## 2.1 RETRAINING APPROACHES

Let us first consider the learning rate scheme used during retraining. The original approach by Han et al. (2015) is commonly referred to as FINE TUNING (FT): suppose we train for $T$ epochs using the learning rate schedule $(\eta_t)_{t \leq T}$ and retrain for $T_{\text{rt}}$ epochs per prune-retrain-cycle, then FT retrains the pruned network for $T_{\text{rt}}$ epochs using a fixed constant learning rate, most commonly $\eta_T$. It was first noticed by Renda et al. (2020) that the precise learning rate schedule during retraining can have a dramatic impact on the predictive performance of the pruned network. Motivated by WEIGHT REWINDING (WR) (Frankle et al., 2019), they proposed LEARNING RATE REWINDING (LRW), where one retrains the pruned network for $T_{\text{rt}}$ epochs using the last $T - T_{\text{rt}}$ learning rates $\eta_{T-T_{\text{rt}}+1}, \dots, \eta_T$. Le & Hua (2021) argued that the reason behind the success of LRW is the usage of large learning rates and proposed SCALED LEARNING RATE RESTARTING (SLR), where the pruned network is retrained using a proportionally identical learning schedule, i.e., by compressing $(\eta_t)_{t \leq T}$ into the retraining time frame of $T_{\text{rt}}$ epochs with a short warm-up phase. They also introduced CYCLIC LEARNING RATE RESTARTING (CLR) based on the the 1-cycle learning rate schedule of Smith & Topin (2017).

We think that the nature of the proposed retraining methods indicates that the retraining phase is, at its core, similar to the usual training phase. Following this rationale, the success of LRW, SLR and CLR over FT should be attributed to the existence of both a large- and small-step retraining regime. In fact, a large initial and exponentially decaying learning rate has become the standard practice for regular training (Leclerc & Madry, 2020). Note that such a scheme is employed not just by SLR and CLR, but also by LRW if $T_{\text{rt}}$ is sufficiently large to model the decaying learning rate schedule of the original training phase. The conventional approach to explaining the success of decaying learning rate schedules comes from an optimization perspective, i.e., an initially large learning rate accelerates training and avoids local minima, while the gradual decay helps to converge to an optimum without oscillation around it. However, an active line of research has theoretically supported the usage of large learning rates and separating training into a large- and small-step regime from a generalization perspective (Jastrzębski et al., 2017; Li et al., 2019; You et al., 2019; Leclerc & Madry, 2020). Put more succinctly: retraining is training and therefore requires that some effort is put into tuning the learning rate scheme. LRW, SLR and CLR provide some good heuristic guidance for how to effectively do so without an insurmountable amount of hyperparameter tuning.

While One Shot IMP, that is IMP with a single prune-retrain cycle, is a viable approach to model pruning, only the iterative approach (with multiple prune-retrain cycles) has been shown to achieve

---

[1]There is another way to view this relation: one can fix a given percentile to be pruned in every iteration and then simply repeat the prune-retrain cycle until either a desired level of sparsity is reached or the performance degradation exceeds a given threshold. This is in fact how it appears to be used by Han et al. (2015) and how it is for example presented by Renda et al. (2020). While this reframing may appear trivial, it in fact highlights a strength of IMP that we will further emphasize when contrasting it with pruning stable approaches.

state-of-the-art accuracy-vs.-sparsity tradeoffs (Han et al., 2015; Renda et al., 2020). This iterative approach however commonly consists of a significant amount of prune-retrain cycles, each needing the full original training time, that is $T = T_{rt}$, resulting in several thousand epochs worth of total training time. Renda et al. (2020) for example suggested the following approach: train a network for $T$ epochs and then iteratively prune 20% percent of the weights and retrain for $T_{rt} = T$ epochs using LRW, i.e., use the same learning rate scheme as during training, until the desired sparsity is reached. We note that for $T = T_{rt}$ the learning rate scheme of SLR becomes essentially identical to that of LRW. For a goal sparsity of 98% and $T = 200$ original training epochs, the algorithm would therefore require 18 prune-retrain-cycles for a massive 3800 total retrain epochs. In Subsection 3.1 we will study the effect of the number of prune-retrain cycles, the number of retraining epochs and the learning rate scheme on the performance of the pruned network to establish whether IMP truly requires this massive amount of computational investment to develop its full potential.

## 2.2 PRUNING SELECTION CRITERIA

IMP in its original form treats all trainable parameters as a single vector and computes a global threshold below which parameters are removed, independent of the layer they belong to. This simple approach, which we will refer to as GLOBAL, has been subject to criticism for not determining optimal layer-dependent pruning rates and for being inconsistent (Liu et al., 2020). Fully-connected layers for example have many more parameters than convolutional layers and are therefore much less sensitive to weight removal (Han et al., 2015; Carreira-Perpiñán & Idelbayev, 2018). Further, it has been observed that the position of a layer can play a role in whether that layer is amenable to pruning: often first and last layers are claimed to be especially relevant for the classification performance (Gale et al., 2019). On the other hand, in which layers pruning takes place significantly impacts the sparsity-induced theoretical speedup (Blalock et al., 2020). Lastly, the non-negative homogeneity of modern ReLU-based neural network architectures (Neyshabur et al., 2015) would also seem to indicate a certain amount of arbitrariness to this heuristic selection rule, or at least a strong dependence on the network initialization rule and optimizer used, as weights can be rescaled to force it to fully remove all parameters of a layer, destroying the pruned network without having affected the output of the unpruned network.

Determining which weights to remove is hence crucial for successful pruning and several methods have been designed to address this fact. Zhu & Gupta (2017) introduced the UNIFORM allocation, in which a global sparsity level is enforced by pruning each layer to exactly this sparsity. Gale et al. (2019) extend this approach in the form of UNIFORM+ by (a) keeping the first convolutional layer dense and (b) pruning at most 80% of the connections in the last fully-connected layer. Evci et al. (2020) propose a reformulation of the ERDŐS-RÉNYI KERNEL (ERK) (Mocanu et al., 2018) to take the layer and kernel dimensions into account when determining the layerwise sparsity distribution. In particular, ERK allocates higher sparsity to layers with more parameters. Finally, Lee et al. (2020) propose LAYER-ADAPTIVE MAGNITUDE-BASED PRUNING (LAMP), an approach which takes an $\ell_2$-distortion perspective by relaxing the problem of minimizing the output distortion at time of pruning with respect to the worst-case input. We note that we follow the advice of Evci et al. (2020) and Dettmers & Zettlemoyer (2019) and do not prune biases and batch-normalization parameters, since they only amount to a negligible fraction of the total weights, however keeping them has a very positive impact on the performance of the learned model.

We will compare these approaches in Subsection 3.2 with a focus on the impact of the retraining phase. Since Le & Hua (2021) found that SLR can be used to obtain strong results even when pruning convolutional filters randomly, i.e., by assigning random importance scores to the filters instead of using the magnitude criterion or others, we are interested in understanding the importance of the retraining technique when considering different sparsity distributions.

## 2.3 A FAIR COMPARISON TO PRUNING STABLE METHODS

Pruning stable algorithms are defined by their ability to find a well-performing pruned model *during the training procedure*. They do so by inducing a strong implicit bias during training, either by gradual pruning, i.e., extending the pruning mask dynamically, or by employing regularization- and constraint-optimization techniques to learn an almost sparse structure throughout training. LC by Carreira-Perpiñán & Idelbayev (2018) and GSM by Ding et al. (2019) both employ a modification

of weight decay and force the $k$ weights with the smallest score more rapidly towards zero, where $k$ is the number of parameters that will eventually be pruned and the score is the parameter magnitude or its product with the loss gradient. Similarly, DNW by Wortsman et al. (2019) zeroes out the smallest $k$ weights in the forward pass while still using a dense gradient. CS by Savarese et al. (2020), STR by Kusupati et al. (2020) and DST (Liu et al., 2020) all rely on the creation of additional trainable threshold parameters, which are applied to sparsify the model while being regularly trained alongside the usual weights. Here, the training objectives are modified via penalty terms to control the sparsification. GMP (Zhu & Gupta, 2017) follows a tunable pruning schedule which sparsifies the network throughout training by dynamically extending and updating a pruning mask. Based on this idea, DPF by Lin et al. (2020) maintains a pruning mask which is extended using the pruning schedule of Zhu & Gupta (2017), but allows for error compensation by modifying the update rule to use the (stochastic) gradient of the pruned model while updating the dense parameters.

The first claimed advantage of pruning stable methods is the lack of a need for retraining, as they produce a sparse and well-performing model at the end of regular training. This however comes at a price: the implicit biases in pruning stable algorithms result in computational overhead when compared to the usual network training, i.e., to find a sparse solution throughout training through extensive regularization, masking or other methods, the per-iteration computational cost is increased. An overview of the computational overhead of the previously introduced methods can be found in Table 3 in the appendix. Neither training epochs nor iterations are therefore an appropriate measure when comparing algorithms, since what matters is the total time required to obtain a sparse network. Gale et al. (2019), Evci et al. (2020) and Lin et al. (2020) consider this when presenting their results. The second claimed advantage of pruning stable methods is that they find a better pruning set than unstable methods that cannot recover mistakenly pruned weights (Liu et al., 2020) and whose heuristic selection mechanism is disadvantaged compared to methods that learn the pruning thresholds (Kusupati et al., 2020). While it seems reasonable to believe that the less performance lost at pruning, the better, Bartoldson et al. (2020) showed that this rationale might not be entirely sound: pruning seems to behave similar to noise injection and there exists a *generalization-stability-tradeoff* indicating that less stability can actually be beneficial to generalization.

Especially given the previously discussed recent advancements in better recovering pruning-induced losses during retraining, both the narrative that the need for retraining is a clear computational disadvantage and that pruning stable methods find better pruning sets stand to be reevaluated. We will do so in Subsection 3.3.

# 3 EXPERIMENTAL RESULTS

Let us outline the general methodological approach to computational experiments in this section, including datasets, architectures, metrics and optimization strategies used. Experiment-specific details are found in the respective subsections. We note that, given the surge of interest in pruning, Blalock et al. (2020) proposed experimental guidelines in the hope of standardizing the experimental setup. We aim to follow these guidelines whenever possible and encourage others to do so as well. All experiments performed throughout this computational study are based on the PyTorch framework (Paszke et al., 2019), using the original code of the methods whenever possible. All results and metrics were logged and analyzed using *Weights & Biases* (Biewald, 2020). We plan on publicly releasing our implementations and general setup for the sake of reproducibility.

We restrict ourselves to the task of image recognition on the publicly available *CIFAR-10* and *CIFAR-100* (Krizhevsky et al., 2009) datasets as well as *ImageNet* (Russakovsky et al., 2015). All experiments are performed using deep convolutional neural network architectures, in particular on *Residual Networks* (He et al., 2015), *Wide Residual Networks* (WRN) (Zagoruyko & Komodakis, 2016) and *VGG-16* (Simonyan & Zisserman, 2014). We use Stochastic Gradient Descent (SGD) with momentum as an optimizer throughout the experiments and exclusively use stepped learning rate schedules. We therefore also decided to forego CLR in favor of SLR despite the recommendations of Le & Hua (2021), though we have included results using CLR in the appendix. Exact parameters can be found in Table 2 in the appendix.

The focus of our analysis will be the tradeoff between the model sparsity and the final test accuracy. As a secondary measure, we will also consider the *theoretical speedup* (Blalock et al., 2020) induced by the sparsity, which is defined as the ratio of the FLOPs needed to evaluate the sparse over the

dense model assuming a theoretical matrix multiplication algorithm capable of taking full advantage of unstructured sparsity, see Subsection A.1 for full details. We decided to include it since it can help to give a more differentiated picture of the inherent tradeoffs.

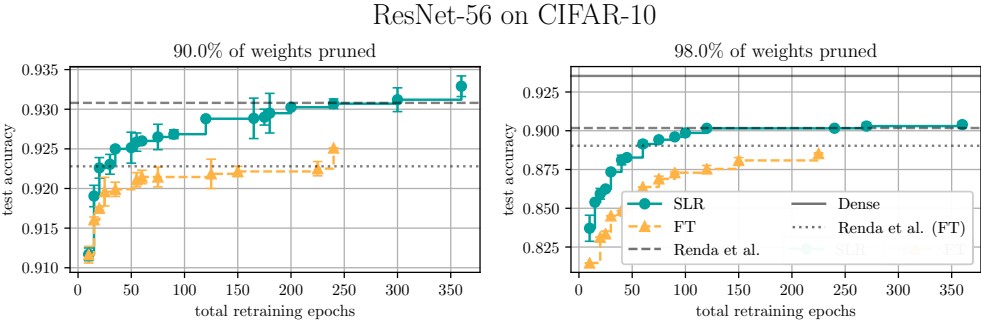

Figure 1: Test accuracy achieved by IMP in relation to the total number of epochs required for retraining using either FT or SLR. For each point on the x-axis, we show the highest mean accuracy achieved by any configuration of a grid search using up to this many total retraining epochs.

### 3.1    THE COMPUTATIONAL COST OF IMP

In this part we will treat the number of retrain epochs per prune-retrain cycle $T_{\mathrm{rt}}$ as well as the total amount of such cycles $J$ as tunable hyperparameters for IMP and try to determine the tradeoff between the predictive performance of the final pruned network and the total number of retrained epochs $J \cdot T_{\mathrm{rt}}$. As a baseline performance for a pruned network, we will use the approach suggested by Renda et al. (2020) as it serves as a good benchmark for the current potential of IMP. We also include a variant of this approach using FT during retraining. We will use the original global pruning criterion of Han et al. (2015) throughout this part.

In Figure 1 we present the results of our computations for ResNet-56 trained on CIFAR-10 with a moderate and high target sparsity of respectively 90% and 98%. The parameters for the retrain phase were optimized using a grid search over $T_{\mathrm{rt}} \in \{10, 15, 20, ..., 60\}$ and $J \in \{1, 2, ..., 6\}$ and we consider both FT and SLR as learning rate schemes during retraining. The weight decay values, including those used for the pruned and unpruned baselines, were individually tuned for each datapoint using a grid search over 1e-4, 2e-4 and 5e-4. All of our results are averaged over 2 seeds with max-min-bands indicated.

Summarizing the results, we find that SLR significantly outperforms FT at both levels of sparsity. While the general efficacy of tuning the learning rate scheme of IMP during retraining has previously been sufficiently demonstrated by Renda et al. (2020) and Le & Hua (2021), the fact that IMP achieves its respective potential with significantly less than the total number of retraining epochs usually budgeted for its full iterative form has to our knowledge not been previously formally established: SLR comes within half a percentage point of the pruned baseline within a mere 100 retraining epochs. This stands in stark contrast to the full 2000 and 3600 retraining epochs required to establish the respective baseline. Considering the pruned baseline using FT during retraining, this fact largely also does not seem to be a particular consequence of the learning rate scheme used during retraining. Unlike commonly assumed, IMP with the right learning rate scheme during retraining therefore seems at least competitive with other methods not just with respect to the predictive power of the pruned network it obtains but also with respect to its total computational cost. We also note that we found IMP with SLR and a shortened retrain phase to benefit from significantly larger weight decay values than the baseline method, see extended results in Subsection B.1 in the appendix.

### 3.2    THE IMPORTANCE OF THE SPARSITY DISTRIBUTION

We compare the original global pruning criterion of IMP to the previously introduced proposed alternatives. Figure 2 reports the test accuracy in relation to the level of sparsity in the One Shot setting for CIFAR-100, where the network is retrained for 30 epochs, and ImageNet, where it is

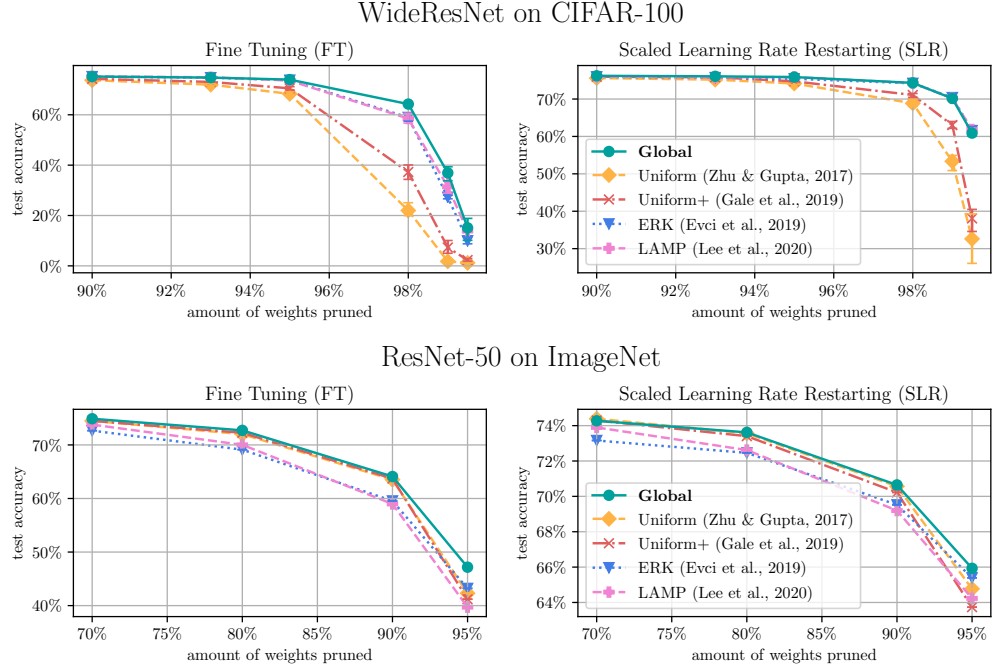

Figure 2: Performance-vs.-sparsity tradeoffs for the One Shot setting on CIFAR-100 (top) and ImageNet (bottom). We compare the sparsity allocation methods w.r.t. the different retraining techniques FT (left) and SLR (right).

retrained for 10 epochs. We tested both FT (Han et al., 2015) and SLR (Le & Hua, 2021) to see if the learning rate scheme during retraining has any impact on the performance of the pruning selection scheme. The CIFAR-100 results are averaged over three seeds and max-min-bands are indicated. Extended plots including the CIFAR-10 dataset as well as the pruning-induced theoretical speedups can be found in Subsection B.2 in the appendix. For ImageNet, the results are based on a single seed.

Surprisingly the simple global selection criterion performs at least on par with the best out of all tested methods at any sparsity level for every combination of dataset and architecture tested here when considering the sparsity of the pruned network as the relevant measure. Using SLR during retraining compresses the results by equalizing performance, but otherwise does not change the overall picture. We note that the results on CIFAR-100 using FT largely track with those reported by Lee et al. (2020), with the exception of the strong performance of the global selection criterion. Apart from slightly different network architectures, we note that they used significantly more retraining epochs, e.g., 100 instead of 30, and that they use AdamW (Loshchilov & Hutter, 2019) instead of SGD. Comparing the impact different optimizers can have on the pruning selection schemes seems like a potentially interesting direction for future research.

While the sparsity-vs.-performance tradeoff has certainly been an important part of the justification of modifications to global selection criterion, let us also directly address two further points that are commonly made in this context. First, the global selection criterion has previously been reported to suffer from a pruning-induced collapse at very high levels of sparsity in certain network architectures that is avoided by other approaches. This phenomenon has been studied in the *pruning before training* literature and was coined *layer-collapse* by Tanaka et al. (2020), who hypothesize that it can be avoided by using smaller pruning steps since gradient descent restabilizes the network after pruning by following a layer-wise magnitude conservation law. To verify whether these observations also hold in the *pruning after training* setting, we trained a VGG-16 network on CIFAR-10, as also done by Lee et al. (2020), both in the One Shot and in the iterative setting. The results are reported in Figure 8 in the appendix and show that layer collapse is clearly occurring for both FT and SLR for the global selection criterion at sparsity levels above 99% in the One Shot setting, but disappears

entirely when pruning iteratively. This indicates that layer collapse, while a genuine potential issue, can be avoided even using the global selection criterion. We also remark that SLR needs less prune-retrain-cycles to avoid layer-collapse than FT, possibly indicating that the retraining strategy impacts the speed of restabilization of the network in the hypothesis posed by Tanaka et al. (2020).

The second important aspect to consider is that layer-dependent selection criteria are also intended to address the inherent tradeoff not just between the achieved sparsity of the pruned network and its performance, but also the theoretical computational speedup. We have included plots highlighting the achieved performance in relation to the theoretical speedup in Subsection B.2 in the appendix. The key takeaway here is that for both the ResNet-56 and the WideResNet network architecture, there is overall surprisingly little distinction between all five tested methods, with Uniform+ and ERK taking the lead and the global selection criterion performing well to average. For the ResNet-50 architecture however a much more drastic separation occurs, with Uniform performing the best, followed by Uniform+ and then the global selection criterion. Overall, the picture is significantly less clear. Our results however indicate that the global selection criterion at the very least seems to offer a good balance in this inherent tradeoff.

Table 1: ResNet-56 on CIFAR-10: Comparison between IMP and pruning stable methods for goal sparsity levels of 90%, 95% and 98%, denoted in the main columns. Each subcolumn denotes the Top-1 accuracy, the theoretical speedup and the actual sparsity achieved by the method. Each row corresponds to one method, where we denote the time needed when compared to regular training next to the method's name. All results include standard deviations where zero or near-zero values are omitted. The two highest accuracy values are highlighted for each sparsity level.

| Method | Time | 90% | | | 95% | | | 98% | | |
|---|---|---|---|---|---|---|---|---|---|---|
| | | Accuracy | Speedup | Sparsity | Accuracy | Speedup | Sparsity | Accuracy | Speedup | Sparsity |
| **IMP** | 1.15x | 92.79 ±0.21 | 6 ±0.4 | 90.00 | 91.62 ±0.29 | 12 ±0.6 | 95.00 | 87.93 ±0.03 | 28 ±1.7 | 98.00 |
| **IMP+** | 1.50x | **93.25 ±0.14** | 6 ±0.4 | 90.00 | **92.57 ±0.18** | 13 ±0.6 | 95.00 | 89.86 ±0.14 | 29 ±1.6 | 98.00 |
| **GMP** | 1.05x | 92.84 ±0.42 | 10 | 90.00 | 92.12 ±0.17 | 20 | 95.00 | 89.65 ±0.31 | 50 | 98.00 |
| **GSM** | 1.17x | 90.83 ±0.24 | 5 | 90.24 | 88.91 ±0.15 | 11 ±0.1 | 95.24 | 85.35 ±0.24 | 23 ±0.9 | 98.24 |
| **DPF** | 1.03x | **93.32 ±0.11** | 7 | 90.00 | **92.68 ±0.14** | 12 ±0.1 | 95.00 | **90.49 ±0.23** | 29 ±1.2 | 98.00 |
| **DNW** | 1.05x | 91.81 ±1.83 | 6 ±0.8 | 90.00 | 91.95 ±0.06 | 7 ±0.3 | 95.09 | 34.87 ±43.08 | 26 ±2.8 | 98.10 |
| **LC** | 1.31x | 90.51 ±0.16 | 5 ±0.1 | 90.00 | 89.16 ±0.60 | 8 ±0.5 | 95.00 | 85.11 ±0.51 | 16 | 98.00 |
| **STR** | 1.35x | 89.25 ±1.23 | 8 ±0.8 | 90.15 ±0.76 | 89.77 ±1.75 | 31 ±10.3 | 95.11 ±0.28 | 89.15 ±0.26 | 66 ±4.9 | 98.00 ±0.04 |
| **CS** | 1.67x | 91.87 ±0.30 | 13 ±0.3 | 90.52 ±0.76 | 91.36 ±0.23 | 21 ±2.9 | 95.38 ±0.19 | **90.04 ±0.36** | 50 ±7.2 | 98.12 ±0.06 |
| **DST** | 2.41x | 92.41 ±0.28 | 10 ±0.7 | 89.55 ±0.41 | 89.17 | 18 | 94.42 | 88.22 ±0.36 | 53 ±3.6 | 98.04 ±0.21 |

## 3.3 COMPARING IMP TO PRUNING STABLE APPROACHES

Using the lessons from the previous two sections, we finally compare IMP to recent pruning stable approaches. Given the high computational demand of tuning ten different methods, we limit ourselves to ResNet-56 networks trained on CIFAR-10 as well as WideResNet for CIFAR-100. For IMP we employ the global selection criterion and retrain using SLR, where the number of prune-retrain-cycles $J$ as well as the number of retraining epochs per cycle $T_{rt}$ were tuned using a grid search over $T_{rt} \in \{10, 15, 20, 25\}$ and $J \in \{1, 2, 3, 4\}$ for CIFAR-10 and over $T_{rt} \in \{5, 10, 15\}$ and $J \in \{1, 2, 3\}$ for CIFAR-100. We will also consider a restricted version of IMP, where we impose the additional constraint $J \cdot T_{rt} \leq 30$ for CIFAR-10 and $J \cdot T_{rt} \leq 20$ for CIFAR-100 to obtain a method that will be largely comparable to pruning stable methods w.r.t. its total runtime. In the following, we denote the restricted version by IMP and the unrestricted version by IMP+.

The hyperparameters of each of the pruning stable methods were likewise tuned using manually defined grid searches, resorting to the recommendations of the original publications whenever possible, see Subsection A.2 for exact details. Whenever possible, that is for GMP, GSM, DNW, DPF, and LC, we give the methods predetermined sparsity levels of 90%, 93%, 95%, 98%, 99% and 99.5%, the same as given to IMP. Tuning the remaining methods in a way that allows for a fair comparison however is significantly more difficult, since none of them allow to clearly specify a desired level of sparsity but instead require one to tune additional hyperparameters as part of the grid search. Despite our best efforts, we were only able to cover part of the desired sparsity range using STR and DST. In addition, we noticed that each of these method can have some variance in the level of sparsity achieved even for fixed hyperparameters, so we list the standard deviation of the final sparsity with

respect to the random seed initialization. All results are averaged over three seeds for CIFAR-10 and over two seeds for CIFAR-100.

One of the main intentions behind pruning stability is to eliminate the need for computationally expensive retraining. We start by verifying this assumption using the pruning stable approaches and allowing them to be retrained for further 30 epochs using FT. Figure 9 and Figure 11 in the appendix show the increase in accuracy of the pruned networks after retraining. We observe that LC and CS are often unable to find an actually sparse model throughout training and hence greatly benefit from retraining. The remaining methods however mostly stay true to their claimed pruning stability, but nevertheless can profit from retraining. For the sake of a fair comparison, we will therefore use the accuracy reached after retraining when it exceeds the original one for all pruning stable methods in this section while only referring to the original train time when comparing runtimes for all methods except LC, CS and of course IMP.

Table 1 now reports the computational cost, final test performance, theoretical speedup and actually achieved sparsity of all ten methods for CIFAR-10. Full results for both CIFAR-10 and CIFAR-100 are reported in Subsection B.3 in the appendix. Regarding the computational costs, we see that only few methods consistently outperform IMP. The results show that for CIFAR-10 only GMP, DPF and DNW deliver a pruned network faster than the restricted form of IMP. While IMP+ is about 30% slower than the restricted form, it still beats CS and DST with respect to its runtime. The network architecture however seems to have a significant impact on the performance of the individual methods, since for CIFAR-100 with a WideResNet architecture, GMP, DPF and STR deliver a pruned network faster than the restricted form of IMP, while IMP+, here about 12% slower than the restricted form, beats GSM, DNW and LC with respect its runtime.

Regarding the achieved accuracy-vs.-sparsity tradeoff, the results show that IMP with SLR and a tuned retrain phase is able to perform on par with the best pruning stable methods considered here. For CIFAR-10, only DPF is able to outperform IMP+ throughout all levels of sparsity, usually by less than 0.3%. All other methods fail to compete at the highest level of sparsity and otherwise at best about match the performance of IMP+. The restricted version of IMP does perform noticeably worse than IMP+ but still compares favorable to most other methods in the comparison throughout all levels of sparsity. For CIFAR-100 however, DPF does not exhibit the same strong behavior despite receiving some additional attention during hyperparameter tuning, see appendix for details. No method is able to consistently outperform even the restricted version of IMP. Somewhat surprisingly, we note that IMP is also able to get favorable theoretical speedups compared to other methods for both CIFAR-10 and CIFAR-100. We emphasize again that IMP manages to obtain these results without a strong computational overhead as commonly assumed.

## 4    DISCUSSION

The goal of this work is to demonstrate that IMP is surprisingly efficient, achieving its full potential with significantly less than the extensive amount of retraining usually considered necessary. We also find that it can provide state-of-the-art pruning results when paired with an appropriate learning rate scheme during retraining, further supporting the results of Renda et al. (2020) and Le & Hua (2021). In our opinion, these results contradict the common narrative in the literature and imply that there is currently insufficient empirical evidence to support the commonly claimed benefits of imposing an implicit bias during training. We hope that our findings establish a more realistic yet easily realisable baseline against which such claims can be compared in the future.

We would also like to acknowledge a clear limitation of the study: our experiments were limited to image recognition tasks on particular network architectures. Likewise, we have limited our exploration to the case of unstructured pruning. Extending the results to include, e.g., NLP as well as a wider variety of network architectures and structured pruning tasks would help paint a more complete picture.

## 5 REPRODUCIBILITY

Reproducibility is of utmost importance for any comparative computational study such as this. All experiments were based on the PyTorch framework and use publicly available datasets. The implementation of the ResNet-56 network architecture is based on `https://github.com/JJGO/shrinkbench/blob/master/models/cifar_resnet.py`, the implementation of the WideResNet network architecture is based on `https://github.com/meliketoy/wide-resnet.pytorch`, the implementation of the VGG-16 network architecture is based on `https://github.com/jaeho-lee/layer-adaptive-sparsity/blob/main/tools/models/vgg.py` and the implementation of the Resnet-50 network architecture is taken from `https://pytorch.org/vision/stable/index.html`. Regarding the pruning methods, the code was taken from the respective publications whenever possible. Regarding the different variants of magnitude pruning such as ERK or Uniform+, we closely followed the implementation of Lee et al. (2020) available at `https://github.com/jaeho-lee/layer-adaptive-sparsity/`. For metrics such as the theoretical speedup, we relied on the implementation in the *ShrinkBench*-framework of Blalock et al. (2020), see `https://github.com/JJGO/shrinkbench`. All parameters considered are specified in Section 3 or in the Appendix. We also plan on publicly releasing the code used for our implementations and general setup for the sake of reproducibility and will make it available during any submission of this work.

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

# A    TECHNICAL DETAILS AND TRAINING SETTINGS

## A.1    TECHNICAL DETAILS

We define *pruning stability* as follows.

**Definition A.1** (Bartoldson et al. (2020))**.** Let $t_{\text{pre}}$ and $t_{\text{post}}$ be the test accuracy before pruning and after pruning the trained model, respectively. We define the *pruning stability* of a method as

$$\Delta_{\text{stability}} := 1 - \frac{t_{\text{pre}} - t_{\text{post}}}{t_{\text{pre}}} \in [0, 1].$$

Pruning stable methods are sparsification algorithms that learn a sparse solution throughout training such that $\Delta_{\text{stability}} \approx 1$. For example, methods that perform the forward-pass using an already sparsified copy of the parameters (e.g. DNW by Wortsman et al., 2019), will have $\Delta_{\text{stability}} = 1$, since the 'hard' pruning step only consists of an application of the present pruning mask, which has no further effect. Methods that actively drive certain parameter groups towards zero more rapidly (such as Carreira-Perpiñán & Idelbayev, 2018; Ding et al., 2019) will have a pruning stability close to 1, since the projection of (magnitude) pruning at the end of training will perturb the parameters only slightly.

Crucial to our analysis are the tradeoffs between the model sparsity, the *final test accuracy* and the *theoretical speedup* induced by the sparsity (Blalock et al., 2020). Theoretical speedup is a metric measuring the ratio in FLOPs needed for inference comparing the dense and sparse model. More precisely, let $F_d$ be the number of FLOPs the dense model needs for inference, and let similarly be $F_s$ the same number for the pruned model, given some sparsity $s$.[2] The theoretical speedup is defined as $F_d/F_s$ and depends solely on the position of the zero weights within the network and layers, not on the numerical values of non-zero parameters. While we focused on the model sparsity in the main body of the text, we include plots regarding the theoretical speedup here in the appendix, as suggested by Blalock et al. (2020).

## A.2    TRAINING SETTINGS AND HYPERPARAMETERS

Table 2: Exact training configurations used throughout the experiments for IMP. For pruning stable methods we use the same settings, with the following exceptions: (1) since momentum plays a crucial part in the justification of GSM, we tune it over 0.9, 0.99 and 0.995 and (2) any additional hyperparameters of the respective methods as well as weight decay were tuned as indicated in Subsection A.2. We note that others have reported an accuracy of around 80% for WRN28x10 trained on CIFAR-100 that we were unable to replicate. The discrepancy is most likely due to an inconsistency in PyTorch's dropout implementation.

| Dataset | Network (number of weights) | Epochs | Batch size | Momentum | Learning rate ($t$ = training epoch) | Unpruned test accuracy |
|---------|------------------------------|--------|------------|----------|--------------------------------------|------------------------|
| CIFAR-10 | ResNet-56 (850 K) VGG-16 (138 Mio) | 200 | 128 | 0.9 | $\eta_t = \begin{cases} 0.1 & t \in [1, 90], \\ 0.01 & t \in [91, 180], \\ 0.001 & t \in [181, 200] \end{cases}$ | 93.5% ±0.3% 93.8% ±0.2% |
| CIFAR-100 | WRN28x10 (37 Mio) | 200 | 128 | 0.9 | $\eta_t = \begin{cases} 0.1 & t \in [1, 60], \\ 0.02 & t \in [61, 120], \\ 0.004 & t \in [121, 160], \\ 0.0008 & t \in [161, 200] \end{cases}$ | 76.7% ±0.2% |
| ImageNet | ResNet-50 (26 Mio) | 90 | 256 | 0.9 | $\eta_t = \begin{cases} 0.1\frac{t}{5} & t \in [1, 5], \\ 0.1 & t \in [5, 30], \\ 0.01 & t \in [31, 60], \\ 0.001 & t \in [61, 80], \\ 0.0001 & t \in [81, 90] \end{cases}$ | 76.17% ±0.03% |

We list the hyperparameter grids used for each pruning stable method taking part in the comparative study.

---

[2]To compute the number of FLOPs, we sample a single batch from the test set. The code to compute the theoretical speedup has been adapted from the repository of the *ShrinkBench* framework (Blalock et al., 2020).

Table 3: Overview of sparsification methods. CS, STR and DST control the sparsity implicitly via additional hyperparameters. IMP is the only method that is pruning instable by design, i.e., it loses its performance right after the ultimate pruning. Further, IMP is the only method that is sparsity agnostic throughout the regular training; the sparsity does not play a role while training to convergence. All other methods require training an entire model when changing the goal sparsity. The computational overhead refers to the per-iteration overhead during regular training. We denote by $n$ the number of trainable parameters, while $k \leq n$ is the number of parameters that remain after pruning to the goal sparsity.

|  | Sparsity specifiable | Pruning stable | Sparsity agnostic training | Overhead during Training |
|---|---|---|---|---|
| IMP | ✓ | ✗ | ✓ | N/A |
| GMP | ✓ | ✓ | ✗ | $\mathcal{O}(n) + \mathcal{O}(n \log n)$ every time the mask is updated |
| GSM | ✓ | ✓ | ✗ | $\mathcal{O}(n + n \cdot \min(k, n-k))$ |
| LC | ✓ | ✓ | ✗ | $\mathcal{O}(n \cdot \min(k, n-k))$ |
| DPF | ✓ | ✓ | ✗ | $\mathcal{O}(n) + \mathcal{O}(n \log n)$ every 16 iterations |
| DNW | ✓ | ✓ | ✗ | $\mathcal{O}(n \cdot \min(k, n-k))$ |
| CS | ✗ | ✓ | ✗ | $\mathcal{O}(n)$ + add. backprop |
| STR | ✗ | ✓ | ✗ | $\mathcal{O}(n)$ + add. backprop |
| DST | ✗ | ✓ | ✗ | $\mathcal{O}(n)$ + add. backprop |

### A.2.1 ResNet-56 on CIFAR-10

For each method (including IMP) we tune weight decay over 1e-4, 5e-4, 1e-3 and 5e-3 and keep momentum fixed at 0.9. Since momentum plays a crucial part in the justification of GSM, we tune it over 0.9, 0.99 and 0.995. Since the learning rate schedule might need additional tuning, we vary the initial learning rate between 0.05, 0.15, 0.1 and 0.2 for all methods except CS. The decay of the schedule follows the same pattern as listed in Table 2. Since CS required the broadest grid, we fixed the learning rate schedule to the one in Table 2. Otherwise, we used the following grids.

**GMP**
Equally distributed pruning steps: $\{20, 100\}$.

**GSM**
Momentum: $\{0.9, 0.95, 0.99\}$.

**LC**
We only tune the weight decay and, similar to the recommendation by Carreira-Perpiñán & Idelbayev (2018), increase it over time as $\lambda_0 \cdot 1.1^j$, where $j$ is increased over time and $\lambda_0$ is the initial weight decay. For the retraining phase we deactivate weight decay.

**DPF**
As for GMP, we tune the number of pruning steps, i.e., $\{20, 100\}$, and the weight decay.

**DNW**
We only tune the weight decay, since there are no additional hyperparameters.

**CS**
As recommended by Savarese et al. (2020), we fix the temperature $\beta$ at 300. Otherwise, we perform the following grid search. We set the mask initialization $s_0 \in \{-0.3, -0.25, -0.2, -0.1, -0.05, 0, 0.05, 0.1, 0.2, 0.25, 0.3\}$ and the $\ell_1$ penalty $\lambda$ to $\{1e\text{-}8, 1e\text{-}7\}$.

**STR**
We tune the initial threshold value $s_{\text{init}} \in \{-100, -50, -5, -2, -1, 0, 5, 50, 100\}$. In an extended grid search, we also used weight decays in $\{5e\text{-}05, 1e\text{-}4\}$ and varied $s_{\text{init}} \in \{-40, -30, -20, -10\}$.

**DST**
We tune the sparsity-controlling regularization parameter $\alpha \in \{5e\text{-}6, 1e\text{-}5, 5e\text{-}5, 1e\text{-}4, 5e\text{-}4\}$. In an extended grid search, we used weight decays in $\{0, 1e\text{-}4\}$ and tuned $\alpha$ over $\{1e\text{-}7, 5e\text{-}7, 1e\text{-}6\}$.

### A.2.2 WIDERESNET ON CIFAR-100

For each method (including IMP) we tune weight decay over 1e-4, 2e-4 and 5e-4 and keep momentum fixed at 0.9. Since momentum plays a crucial part in the justification of GSM, we tune it over 0.9 and 0.95. For CIFAR-10, we noticed that especially DPF, DST and GMP benefited from an additional tuning of the learning rate schedule. We hence vary the initial learning rate between 0.05, 0.15 and 0.1 for these three methods and also for CS to broaden the grid which resulted in inferior performance compared to CS on CIFAR-10. The decay of the schedule follows the same pattern as listed in Table 2. Otherwise, we used the following grids.

**GMP**
Equally distributed pruning steps: $\{20, 100\}$.

**GSM**
Momentum: $\{0.9, 0.95, 0.99\}$.

**LC**
We only tune the weight decay and, similar to the recommendation by Carreira-Perpiñán & Idelbayev (2018), increase it over time as $\lambda_0 \cdot 1.1^j$, where $j$ is increased over time and $\lambda_0$ is the initial weight decay. For the retraining phase we deactivate weight decay.

**DPF**
As for GMP, we tune the number of pruning steps, i.e., $\{20, 100\}$, and the weight decay.

**DNW**
We only tune the weight decay, since there are no additional hyperparameters.

**CS**
As recommended by Savarese et al. (2020), we fixed the temperature $\beta$ at 300, but increased it to 500 upon noticing the pruning instability of CS on this dataset. Otherwise, we perform the following grid search. We set the mask initialization $s_0 \in \{-0.3, -0.25, -0.2, -0.1, -0.05, -0.03, -0.01, -0.005, -0.003, -0.001, 0\}$ and the $\ell_1$ penalty $\lambda$ to $\{1e-9, 1e-8, 1e-7\}$.

**STR**
We tune the initial threshold value $s_{init} \in \{-5000, -3000, -2000, -1000, -500\}$. In an extended grid search, we also varied $s_{init} \in \{-200, -150, -100, -80, -50, -40, -25, -10, 0\}$.

**DST**
We tune the sparsity-controlling regularization parameter $\alpha \in \{5e-6, 1e-5, 5e-5, 1e-4, 5e-4\}$. In an extended grid search, we tuned $\alpha$ over $\{1e-6, 3e-6, 8e-6, 3e-5, 8e-5, 2e-4, 3e-4, 4e-4, 5e-4, 6e-4, 7e-4, 8e-4, 9e-4, 1e-3\}$.

# B ADDITIONAL PLOTS

## B.1 THE COMPUTATIONAL COST OF IMP

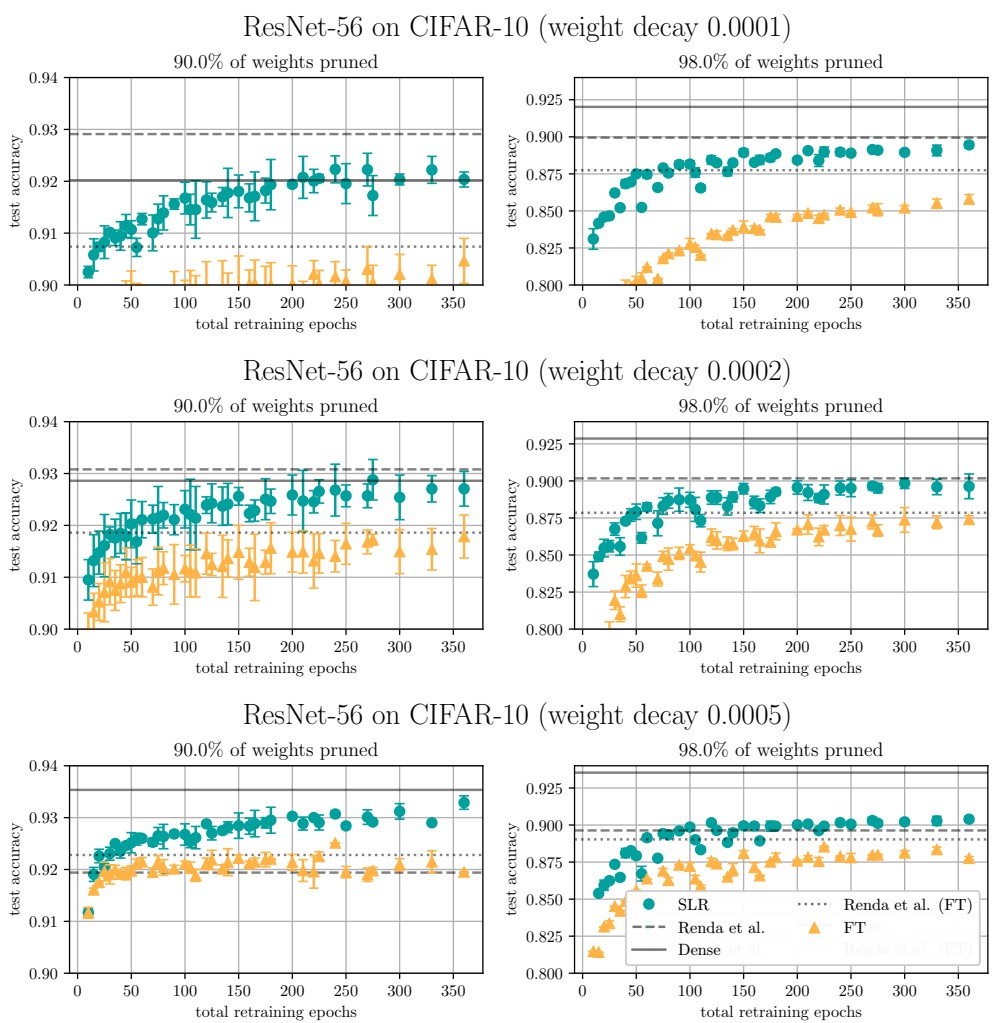

Figure 3: Test accuracy achieved by IMP in relation to the total number of epochs required for retraining using either FT or SLR. Each row of the plots shows a different weight decay as indicated in the title. The pruned baselines were established by using the approach of Renda et al. (2020) using the respective weight decay and either in its original form, using LRW, or with a modified version using FT..

The plots of Figure 3 show the results of Figure 1 for each of the three weight decay factors 1e-4, 2e-4 and 5e-4 independently. Clearly, the choice of the weight decay affects both the accuracy of the dense model as well as the accuracy of the pruned model when using the approach by Renda et al. (2020), both with LRW and FT. For the dense models with weight decays 1e-4, 2e-4 and 5e-4 we obtain respective average performances of 92.01% (±0.41%), 92.86% (±0.49%) and 93.53% (±0.26%). For the respective results of the algorithm by Renda et al. (2020) we observe test accuracies of 92.91% (±0.39%), 93.08% (±0.54%) and 91.94% (±0.16%) in the case of a goal sparsity of 90%, while the results at a goal sparsity of 98% are 89.95% (±0.23%), 90.18% (±0.89%) and 89.63% (±0.16%). Similarly, for the algorithm by Renda et al. (2020) with FT instead of LRW, we observe test accuracies of 90.74% (±0.43%), 91.86% (±0.54%) and 92.28% (±0.02%) in the case of a goal sparsity of 90%, while the results at a goal sparsity of 98% are 87.75% (±0.4%), 87.85% (±0.58%) and 89.03% (±0.09%).

Clearly, we see that SLR benefits from a larger weight decay, while the approach by Renda et al. (2020) is suffering from an increased penalty on the weights. Although SLR is not able to reach the pruned baseline in the case of a 1e-4 weight decay within the given retraining time frame, we note that SLR easily outperforms the LRW-based proposal by Renda et al. (2020) when considering the weight decays that also lead to the best performing dense model, which is a strong indicator that it is preferable to use SLR and a shortened retraining timeframe.

## B.2 THE IMPORTANCE OF THE SPARSITY DISTRIBUTION

In the case of ResNet-56 on CIFAR-10 and VGG-16 on CIFAR-10, we report the weight decay config with highest accuracy, where we optimized over the values 1e-4, 5e-4 and 1e-3. For WideResNet on CIFAR-100 and ResNet-50 on ImageNet we relied on a weight decay value of 1e-4 for both architectures.

Figure 4, Figure 5 and Figure 6 compare the different sparsity allocation methods with respect to the retraining strategy for ResNet-56 on CIFAR-10, WideResNet on CIFAR-100 and ResNet-50 on ImageNet. Similarly, Figure 7 shows the accuracy vs. theoretical speedup tradeoffs on a logarithmic scale. These plots also show that some methods sparsify convolutional layers more aggressively than the global approach, resulting in higher theoretical speedups. However, despite its simplicity, the global approach performs on par with respect to managing the accuracy vs. speedup tradeoff, where we observe that for ResNet-50 on ImageNet it even outperforms methods such as LAMP and ERK regarding both objectives, performance and speedup.

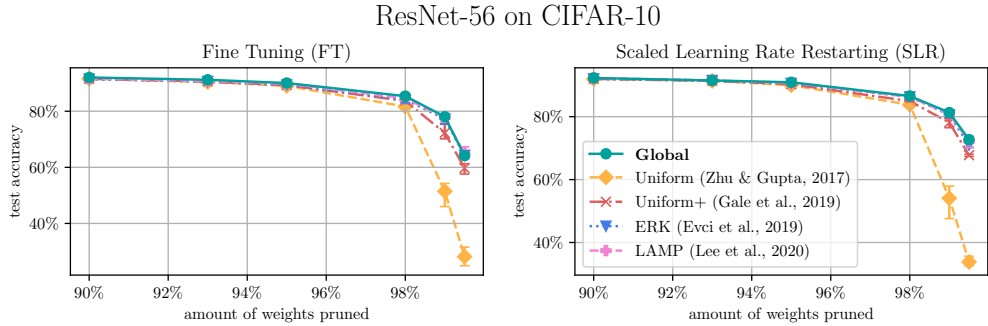

Figure 4: Sparsity-vs.-performance tradeoffs for ResNet-56 on CIFAR-10 for IMP in the One Shot setting for FT (left) and SLR (right) as retraining methods. The plot includes max-min confidence intervals.

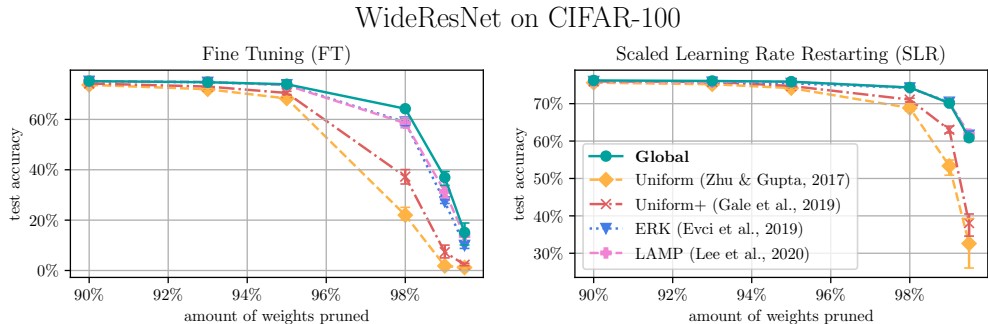

Figure 5: Sparsity-vs.-performance tradeoffs for WideResNet on CIFAR-100 for IMP in the One Shot setting for FT (left) and SLR (right) as retraining methods. The plot includes max-min confidence intervals.

ResNet-50 on ImageNet

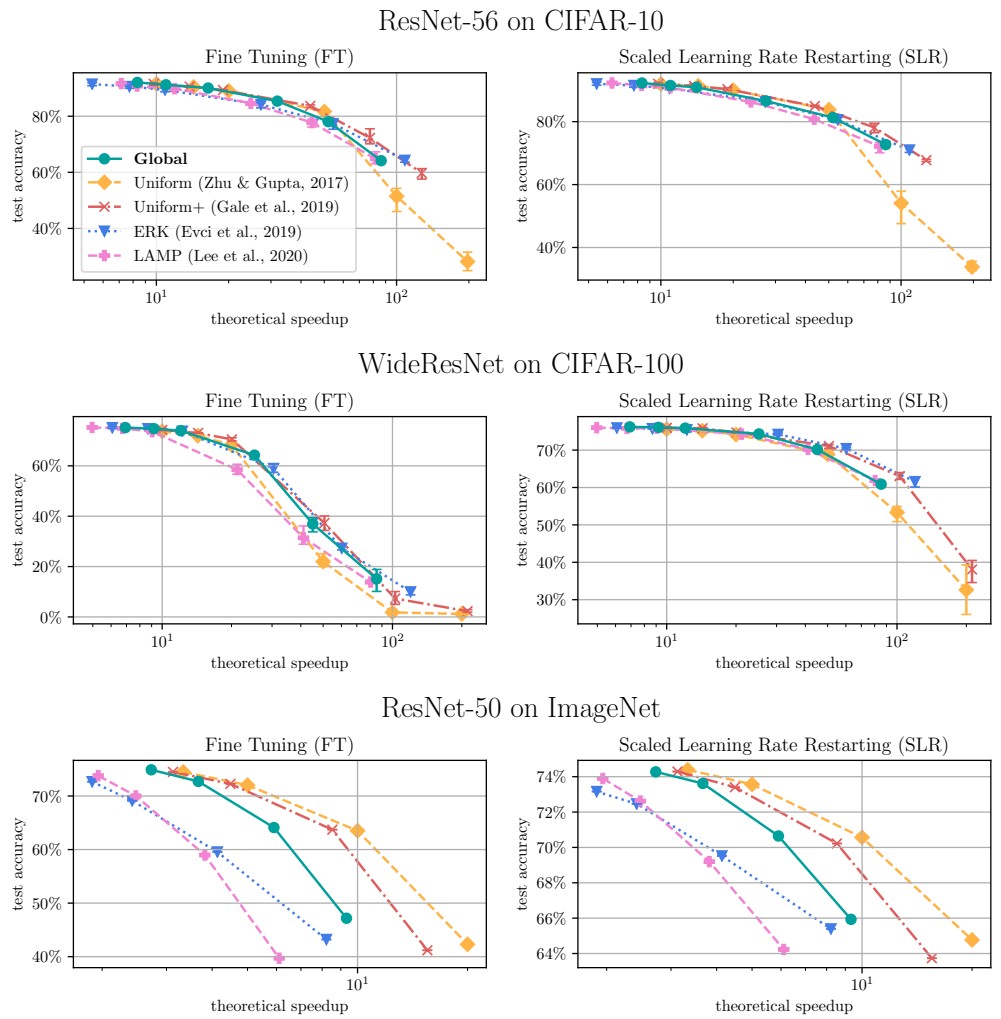

Figure 6: Sparsity-vs.-performance tradeoffs for ResNet-50 on ImageNet for IMP in the One Shot setting for FT (left) and SLR (right) as retraining methods.

Figure 7: Performance-vs.-theoretical speedup tradeoffs for ResNet-56, WideResNet and ResNet-50 on the respective datasets. All plots depict the one shot setting.

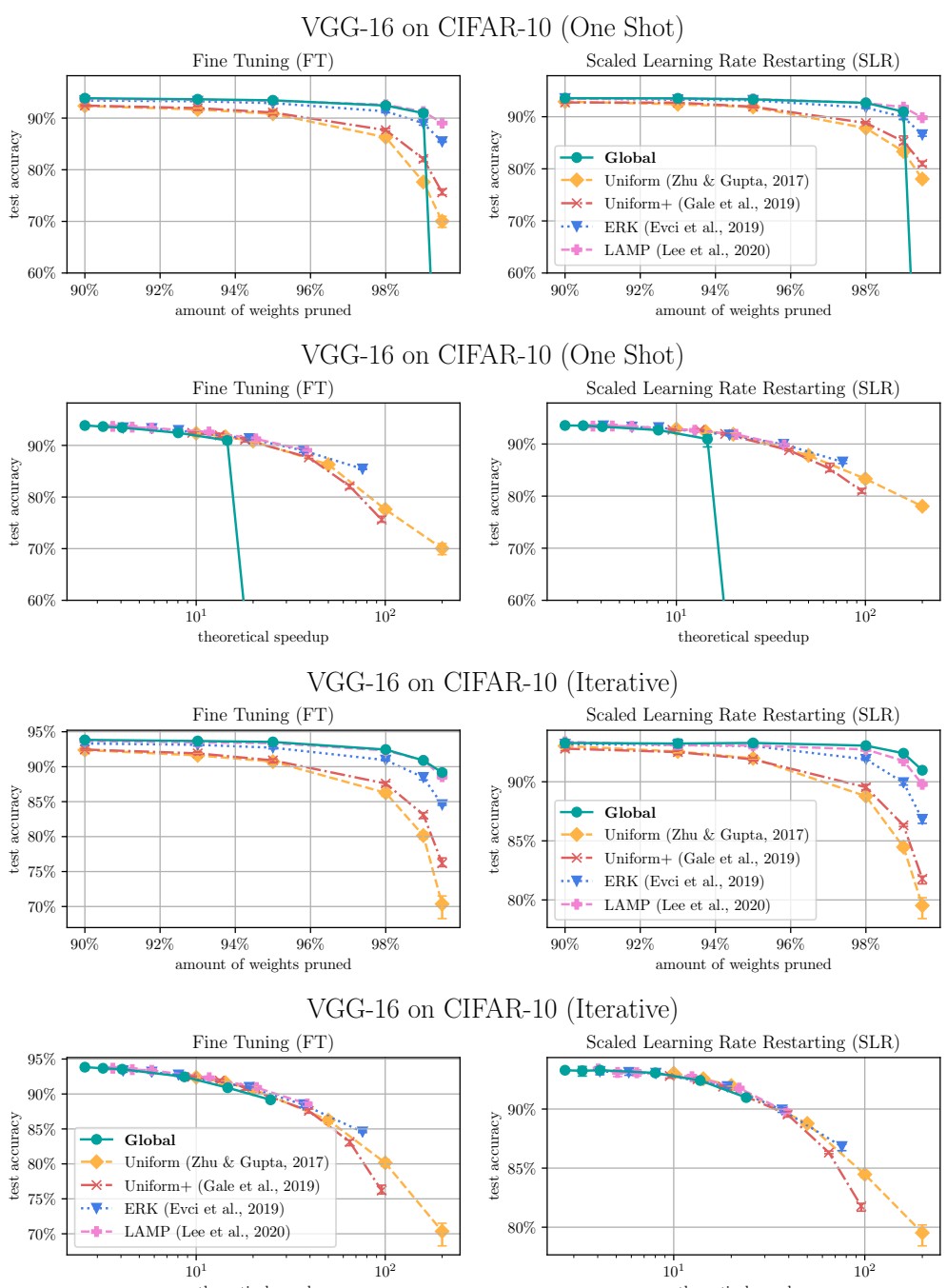

Figure 8: Performance-vs.-sparsity and performance-vs.-theoretical-speedup tradeoffs for VGG-16 on CIFAR-10 for IMP in the One Shot (above) and Iterative (below) setting for FT (left) and SLR (right) as retraining methods. In the One Shot setting the model is retrained for 30 epochs after pruning and the iterative setting consists of 3 prune-retrain cycles with 10 epochs each. For One Shot we observe *layer-collapse* while the iterative splitting into less severe pruning steps avoids the problem. Note that the total amount of retraining epochs between the two settings is identical here.

### B.3 Comparing IMP to Pruning Stable Approaches

Table 4 extends Table 1 by showing the results of the full sparsity range between 90% and 99.5%. The same results can be seen visualized in Figure 10. As described in the main section, Figure 9 shows the actual pruning stability and increase in accuracy after retraining with FT. Apart from LC, all methods can be considered pruning stable with a pruning stability close to 100%. However, we note that some methods can benefit from retraining. To allow a fair comparison, we hence always considered the maximum of the performances before and after retraining, while measuring the computational time needed only for the regular training time, ignoring the time needed for retraining.

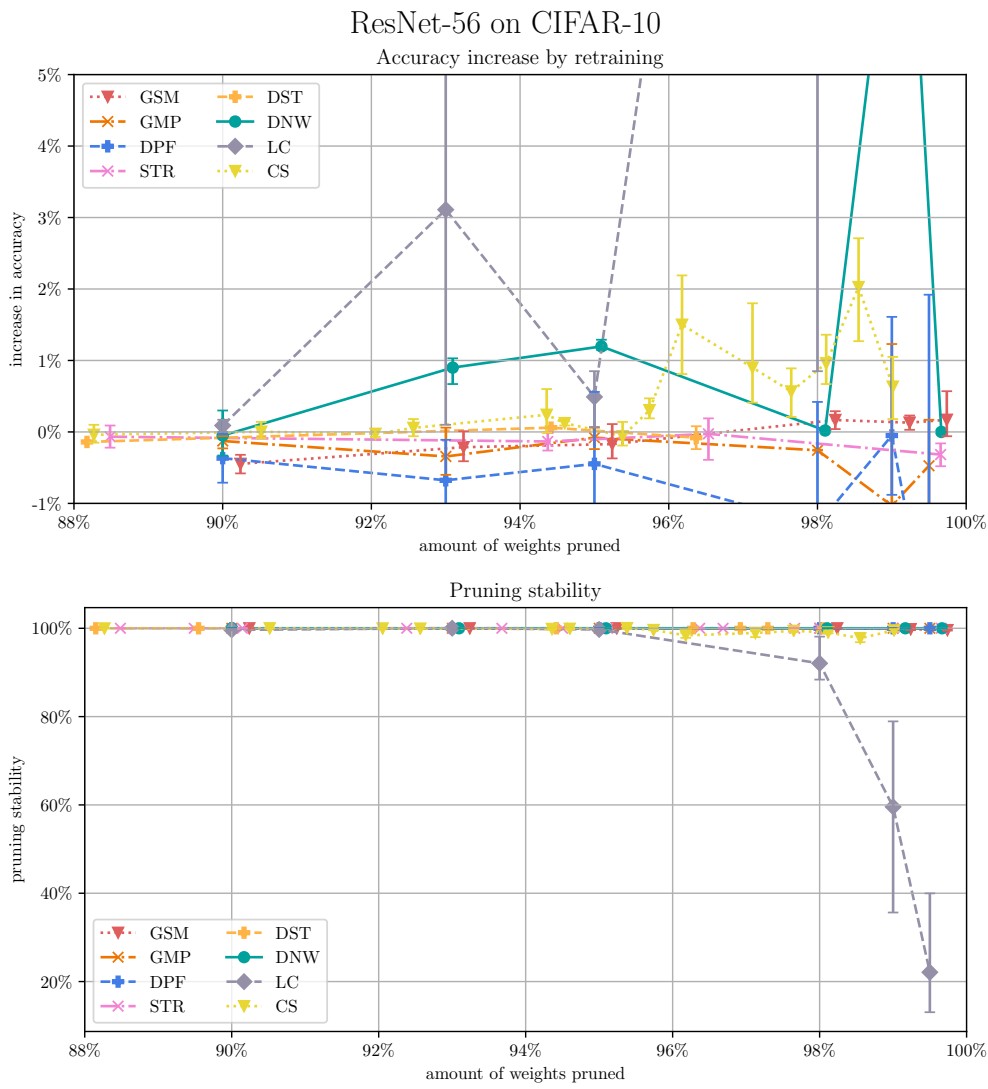

Figure 9: Increase in accuracy after retraining (above) as well as pruning stability (below) for ResNet-56 trained on CIFAR-10 using different pruning stable methods. Each method was retrained for 30 epochs using FT. Each datapoint corresponds to the hyperparameter config with highest accuracy directly after pruning when considering .5% sparsity intervals between 88% and 100%. The confidence bands indicate the min-max-deviation around the mean with respect to different initialization seeds.

Table 4: ResNet-56 on CIFAR-10: Results of the comparison between IMP and pruning stable methods for the sparsity range between 90% and 99.5%. The columns are structured as follows: First the method is stated, where IMP+ denotes the unrestricted version of IMP. Secondly, we denote the time needed when compared to regular training of a dense model, e.g. LC needs 1.14 times as much runtime as regular training. The following columns are substructured as follows: Each column corresponds to one goal sparsity and each subcolumn denotes the Top-1 accuracy, the theoretical speedup and the actual sparsity reached. All results include standard deviations, where we omit zero or close to zero results. Missing values (indicated by —) correspond to cases where we were unable to obtain results in the desired sparsity range, i.e., there did not exist a training configuration with average final sparsity within a .25% interval around the goal sparsity and the closest one is too far away or belongs to another column.

| Method | Time | 90% | | | 93% | | |
|--------|------|----------|---------|----------|----------|---------|----------|
| | | Accuracy | Speedup | Sparsity | Accuracy | Speedup | Sparsity |
| **IMP** | 1.15x | 92.79 ±0.21 | 6 ±0.4 | 90.00 | 92.18 ±0.15 | 9 ±0.5 | 93.00 |
| **IMP+** | 1.50x | **93.25 ±0.14** | 6 ±0.4 | 90.00 | **92.89 ±0.13** | 9 ±0.5 | 93.00 |
| **GMP** | 1.05x | 92.84 ±0.42 | 10 | 90.00 | 92.50 ±0.10 | 14 | 93.00 |
| **GSM** | 1.17x | 90.83 ±0.24 | 5 | 90.24 | 89.92 ±0.14 | 8 ±0.2 | 93.24 |
| **DPF** | 1.03x | **93.32 ±0.11** | 7 | 90.00 | **93.23 ±0.05** | 9 ±0.5 | 93.00 |
| **DNW** | 1.05x | 91.81 ±1.83 | 6 ±0.8 | 90.00 | 91.97 ±0.20 | 5 ±0.2 | 93.09 |
| **LC** | 1.31x | 90.51 ±0.16 | 5 ±0.1 | 90.00 | 89.67 ±0.55 | 7 ±0.6 | 93.00 |
| **STR** | 1.35x | 89.25 ±1.23 | 8 ±0.8 | 90.15 ±0.76 | 90.27 ±0.86 | 20 ±4.0 | 93.01 ±0.58 |
| **CS** | 1.67x | 91.87 ±0.30 | 13 ±0.3 | 90.52 ±0.76 | 89.64 ±1.43 | 16 ±2.9 | 92.78 ±0.18 |
| **DST** | 2.41x | 92.41 ±0.28 | 10 ±0.7 | 89.55 ±0.41 | — | — | — |

| Method | Time | 95% | | | 98% | | |
|--------|------|----------|---------|----------|----------|---------|----------|
| | | Accuracy | Speedup | Sparsity | Accuracy | Speedup | Sparsity |
| **IMP** | 1.15x | 91.62 ±0.29 | 12 ±0.6 | 95.00 | 87.93 ±0.03 | 28 ±1.7 | 98.00 |
| **IMP+** | 1.50x | **92.57 ±0.18** | 13 ±0.6 | 95.00 | 89.86 ±0.14 | 29 ±1.6 | 98.00 |
| **GMP** | 1.05x | 92.12 ±0.17 | 20 | 95.00 | 89.65 ±0.31 | 50 | 98.00 |
| **GSM** | 1.17x | 88.91 ±0.15 | 11 ±0.1 | 95.24 | 85.35 ±0.24 | 23 ±0.9 | 98.24 |
| **DPF** | 1.03x | **92.68 ±0.14** | 12 ±0.1 | 95.00 | **90.49 ±0.23** | 29 ±1.2 | 98.00 |
| **DNW** | 1.05x | 91.95 ±0.06 | 7 ±0.3 | 95.09 | 34.87 ±43.08 | 26 ±2.8 | 98.10 |
| **LC** | 1.31x | 89.16 ±0.60 | 8 ±0.5 | 95.00 | 85.11 ±0.51 | 16 | 98.00 |
| **STR** | 1.35x | 89.77 ±1.75 | 31 ±10.3 | 95.11 ±0.28 | 89.15 ±0.26 | 66 ±4.9 | 98.00 ±0.04 |
| **CS** | 1.67x | 91.36 ±0.23 | 21 ±2.9 | 95.38 ±0.19 | **90.04 ±0.36** | 50 ±7.2 | 98.12 ±0.06 |
| **DST** | 2.41x | 89.17 | 18 | 94.42 | 88.22 ±0.36 | 53 ±3.6 | 98.04 ±0.21 |

| Method | Time | 99% | | | 99.5% | | |
|--------|------|----------|---------|----------|----------|---------|----------|
| | | Accuracy | Speedup | Sparsity | Accuracy | Speedup | Sparsity |
| **IMP** | 1.15x | 83.47 ±0.22 | 51 ±2.4 | 99.00 | 76.09 ±0.20 | 94 ±5.3 | 99.50 |
| **IMP+** | 1.50x | 86.03 ±0.31 | 52 ±2.7 | 99.00 | **79.81 ±0.68** | 87 ±5.8 | 99.50 |
| **GMP** | 1.05x | 62.80 ±1.11 | 100 | 99.00 | 36.95 ±0.32 | 197 | 99.50 |
| **GSM** | 1.17x | 81.09 ±0.04 | 36 ±2.2 | 99.24 | 74.54 ±1.01 | 65 ±4.3 | 99.74 |
| **DPF** | 1.03x | **86.76 ±0.33** | 63 ±3.7 | 99.00 | **80.03 ±0.64** | 146 ±34.3 | 99.50 |
| **DNW** | 1.05x | 83.67 ±0.24 | 15 ±0.1 | 99.17 | 34.71 ±24.17 | 34 ±4.4 | 99.67 |
| **LC** | 1.31x | 81.63 ±0.74 | 30 ±1.5 | 99.00 | 74.44 ±2.03 | 64 ±4.5 | 99.50 |
| **STR** | 1.35x | 83.68 ±0.94 | 159 ±31.9 | 99.13 ±0.02 | 77.34 ±2.68 | 420 ±167.4 | 99.66 ±0.09 |
| **CS** | 1.67x | 86.55 ±0.92 | 69 ±7.1 | 98.90 ±0.02 | — | — | — |
| **DST** | 2.41x | **86.99** | 63 | 98.36 | — | — | — |

## B.4 RESULTS USING CLR

Although our focus lies on analyzing SLR, as explained in Section 3, we include results using CLR in this section: Figure 13 complements Figure 3 and Figure 14 as well as Figure 14 show the effect of CLR onto the pruning selection criteria of interest throughout this work. We do not observe significant differences between SLR and CLR.

Table 5: WideResNet on CIFAR-100: Results of the comparison between IMP and pruning stable methods for the sparsity range between 90% and 99.5%. The columns are structured as follows: First the method is stated, where IMP+ denotes the unrestricted version of IMP. Secondly, we denote the time needed when compared to regular training of a dense model, e.g. LC needs 1.14 times as much runtime as regular training. The following columns are substructured as follows: Each column corresponds to one goal sparsity and each subcolumn denotes the Top-1 accuracy, the theoretical speedup and the actual sparsity reached. All results include standard deviations, where we omit zero or close to zero results. Missing values (indicated by —) correspond to cases where we were unable to obtain results in the desired sparsity range, i.e., there did not exist a training configuration with average final sparsity within a .25% interval around the goal sparsity and the closest one is too far away or belongs to another column.

| | | 90% | | | 93% | | |
|---|---|---|---|---|---|---|---|
| Method | Time | Accuracy | Speedup | Sparsity | Accuracy | Speedup | Sparsity |
| **IMP** | 1.10x | **77.75 ±0.37** | 8 | 90.00 | **77.61 ±0.07** | 11 ±0.2 | 93.00 |
| **IMP+** | 1.23x | 77.72 ±0.47 | 8 | 90.00 | **77.85 ±0.37** | 11 ±0.2 | 93.00 |
| **GMP** | 1.00x | 75.84 ±0.25 | 10 | 90.00 | 75.09 ±0.47 | 14 | 93.00 |
| **GSM** | 2.08x | 74.46 ±0.35 | 6 | 90.02 | 74.34 ±0.39 | 9 | 93.02 |
| **DPF** | 1.08x | 76.74 ±0.22 | 7 ±0.1 | 90.00 | 76.45 ±0.22 | 10 ±0.2 | 93.00 |
| **DNW** | 1.94x | **77.73 ±0.29** | 7 | 90.00 | 76.74 ±0.05 | 7 | 93.00 |
| **LC** | 2.17x | 74.45 ±0.30 | 4 | 90.00 | 73.23 ±0.67 | 6 ±0.1 | 93.00 |
| **STR** | 1.05x | 73.06 ±1.22 | 15 | 90.97 | 74.11 ±0.21 | 13 ±0.2 | 92.36 ±0.17 |
| **CS** | 1.21x | 73.50 ±0.47 | 7 | 89.82 ±0.09 | 73.52 ±0.21 | 10 | 92.96 ±0.03 |
| **DST** | 1.14x | 18.81 ±2.05 | 20 ±0.4 | 90.31 ±0.08 | 66.62 ±1.68 | 23 ±1.4 | 92.89 ±0.11 |
| | | 95% | | | 98% | | |
| Method | Time | Accuracy | Speedup | Sparsity | Accuracy | Speedup | Sparsity |
| **IMP** | 1.10x | **77.56 ±0.22** | 15 ±0.2 | 95.00 | **75.93 ±0.33** | 34 ±0.3 | 98.00 |
| **IMP+** | 1.23x | **77.59 ±0.34** | 15 ±0.3 | 95.00 | **76.27 ±0.23** | 35 ±0.5 | 98.00 |
| **GMP** | 1.00x | 74.52 ±0.19 | 19 | 95.00 | 66.25 ±1.31 | 49 | 98.00 |
| **GSM** | 2.08x | 73.47 ±0.25 | 15 | 95.02 | 70.28 ±0.95 | 23 ±0.1 | 98.02 |
| **DPF** | 1.08x | 75.73 ±0.28 | 16 ±0.4 | 95.00 | 73.05 | 29 | 98.00 |
| **DNW** | 1.94x | 75.69 ±0.21 | 8 | 95.00 | 72.65 ±0.28 | 17 | 98.01 |
| **LC** | 2.17x | 72.41 ±0.54 | 8 ±0.2 | 95.00 | 70.06 ±0.06 | 17 ±0.5 | 98.00 |
| **STR** | 1.05x | 70.66 ±0.52 | 24 | 94.40 ±0.02 | 62.24 ±2.50 | 61 ±0.5 | 97.85 |
| **CS** | 1.21x | 72.81 ±0.13 | 12 | 95.22 | 72.29 ±0.34 | 28 ±0.5 | 97.99 |
| **DST** | 1.14x | 69.33 ±0.28 | 25 ±9.8 | 95.22 ±1.12 | 68.46 | 40 | 97.96 |
| | | 99% | | | 99.5% | | |
| Method | Time | Accuracy | Speedup | Sparsity | Accuracy | Speedup | Sparsity |
| **IMP** | 1.10x | **72.03 ±0.52** | 65 ±1.3 | 99.00 | 65.29 ±0.87 | 130 ±2.3 | 99.50 |
| **IMP+** | 1.23x | **73.72 ±0.15** | 71 ±0.9 | 99.00 | **67.61 ±1.38** | 140 ±1.4 | 99.50 |
| **GMP** | 1.00x | 36.33 ±9.76 | 100 | 99.00 | 16.19 ±7.59 | 200 | 99.50 |
| **GSM** | 2.08x | 66.14 ±0.32 | 32 | 99.02 | 59.64 ±1.35 | 62 ±1.3 | 99.52 |
| **DPF** | 1.08x | 70.82 ±0.64 | 59 ±0.4 | 99.00 | 60.23 ±1.61 | 143 ±3.8 | 99.50 |
| **DNW** | 1.94x | 67.42 ±0.84 | 21 ±0.8 | 99.02 | 62.10 ±1.16 | 42 | 99.52 |
| **LC** | 2.17x | 65.58 ±0.69 | 33 ±0.3 | 99.00 | 59.60 ±0.32 | 74 ±0.8 | 99.50 |
| **STR** | 1.05x | 17.32 ±5.67 | 142 ±2.9 | 98.79 | 10.04 ±0.88 | 271 ±8.5 | 99.33 ±0.01 |
| **CS** | 1.21x | 70.61 ±0.14 | 41 ±0.3 | 98.75 | **68.27 ±0.62** | 82 ±2.6 | 99.40 |
| **DST** | 1.14x | 48.75 | 61 | 98.42 | — | — | — |

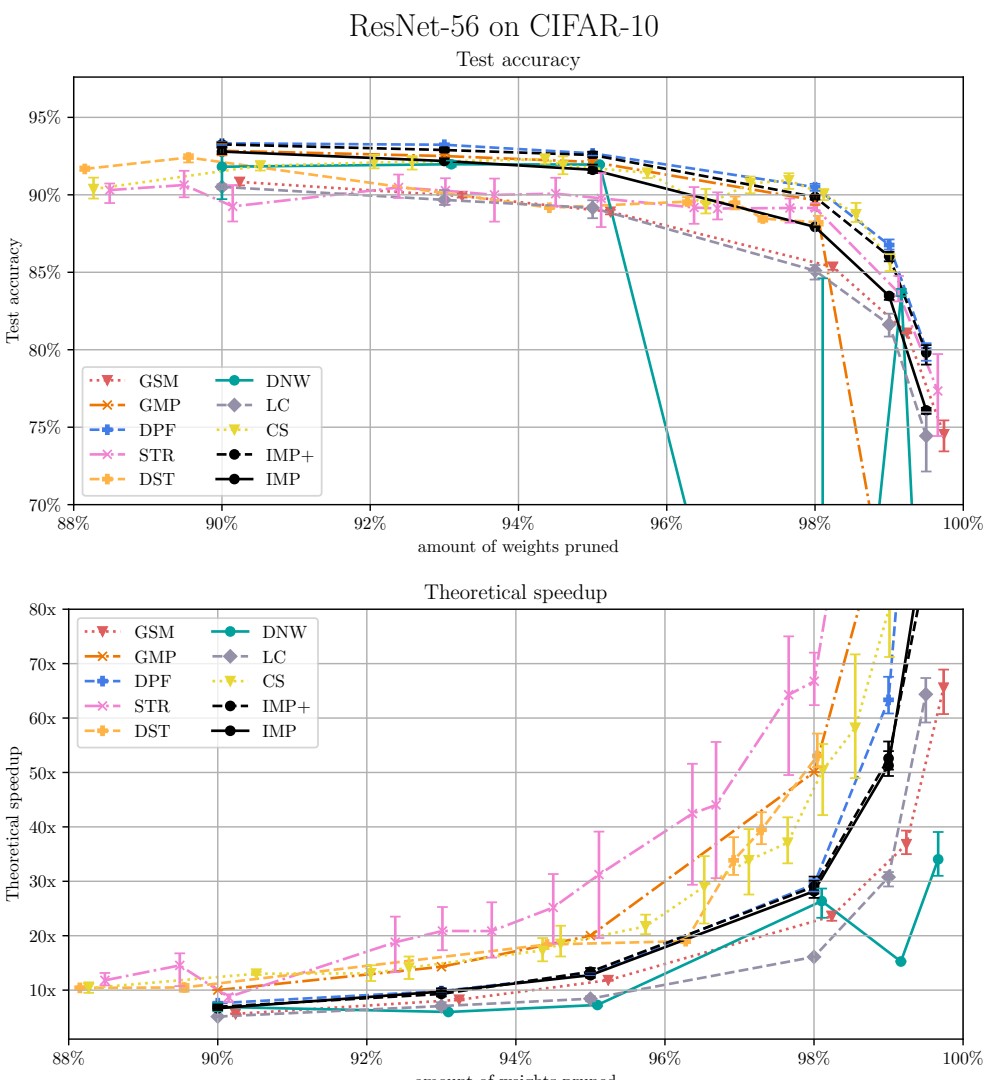

Figure 10: Test accuracy (above) and theoretical speedup (below) of IMP in comparison to different pruning stable methods when training a ResNet-56 network on CIFAR-10. All methods were trained to achieve sparsity levels of 90%, 93%, 95%, 98%, 99% and 99.5% with the exception of CS, STR and DST, where additional hyperparameter searches were necessary to obtain the curves shown here. Each datapoint corresponds to the hyperparameter config with highest accuracy when considering .5% sparsity intervals between 88% and 100%. This holds similarly for the theoretical speedup, where points are selected by highest accuracy as well. The confidence bands indicate the min-max-deviation around the mean with respect to different initialization seeds.

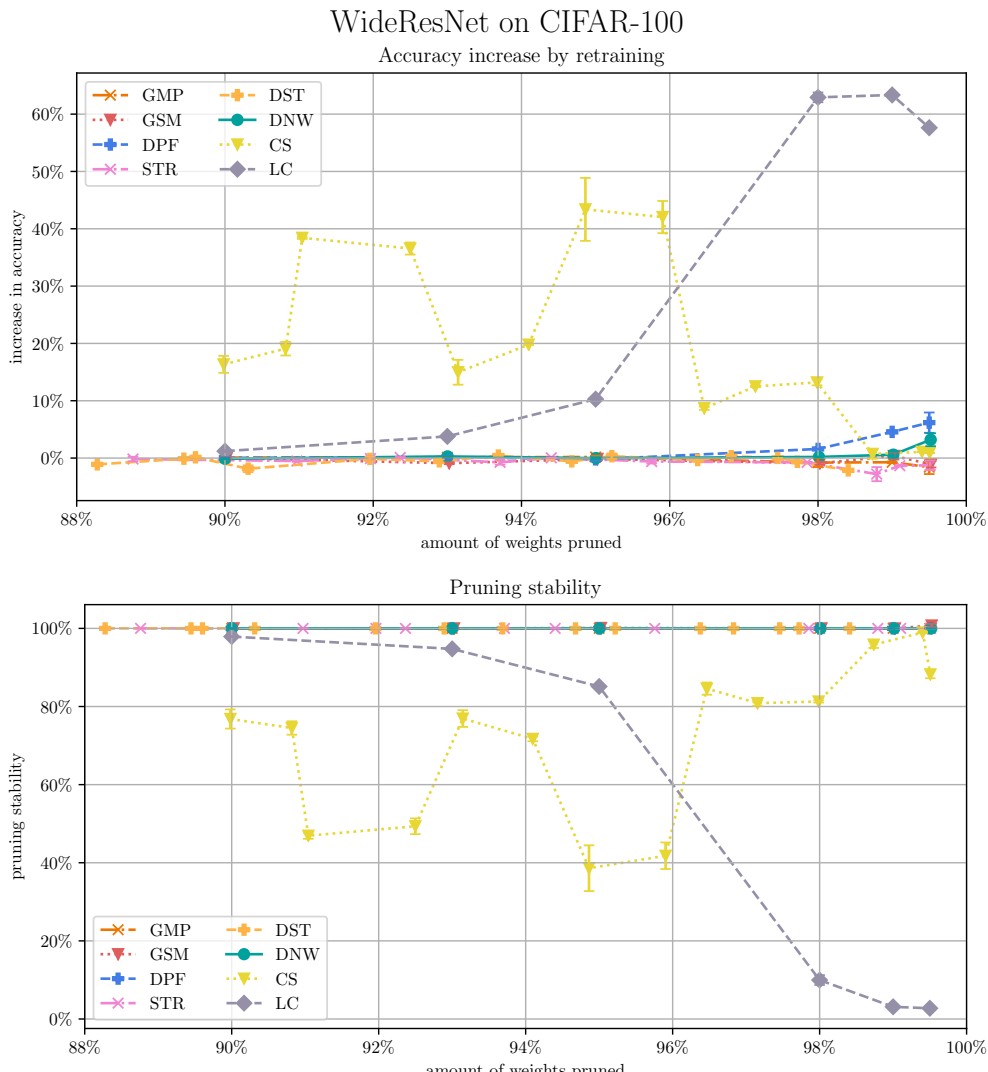

Figure 11: Increase in accuracy after retraining (above) as well as pruning stability (below) for WideResNet trained on CIFAR-100 using different pruning stable methods. Each method was retrained for 30 epochs using FT. Each datapoint corresponds to the hyperparameter config with highest accuracy directly after pruning when considering .5% sparsity intervals between 88% and 100%. The confidence bands indicate the min-max-deviation around the mean with respect to different initialization seeds.

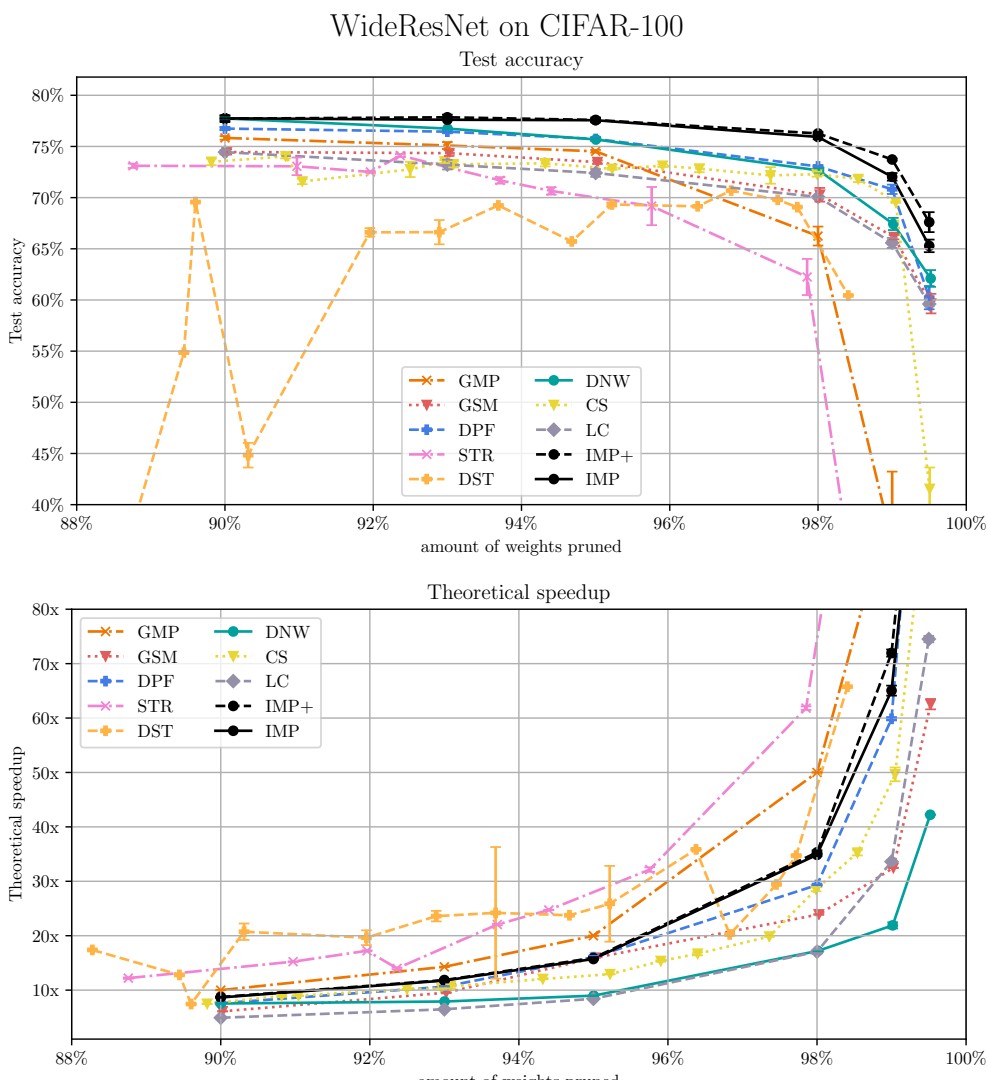

Figure 12: Test accuracy (above) and theoretical speedup (below) of IMP in comparison to different pruning stable methods when training a WideResNet network on CIFAR-100. All methods were trained to achieve sparsity levels of 90%, 93%, 95%, 98%, 99% and 99.5% with the exception of CS, STR and DST, where additional hyperparameter searches were necessary to obtain the curves shown here. Each datapoint corresponds to the hyperparameter config with highest accuracy when considering .5% sparsity intervals between 88% and 100%. This holds similarly for the theoretical speedup, where points are selected by highest accuracy as well. The confidence bands indicate the min-max-deviation around the mean with respect to different initialization seeds.

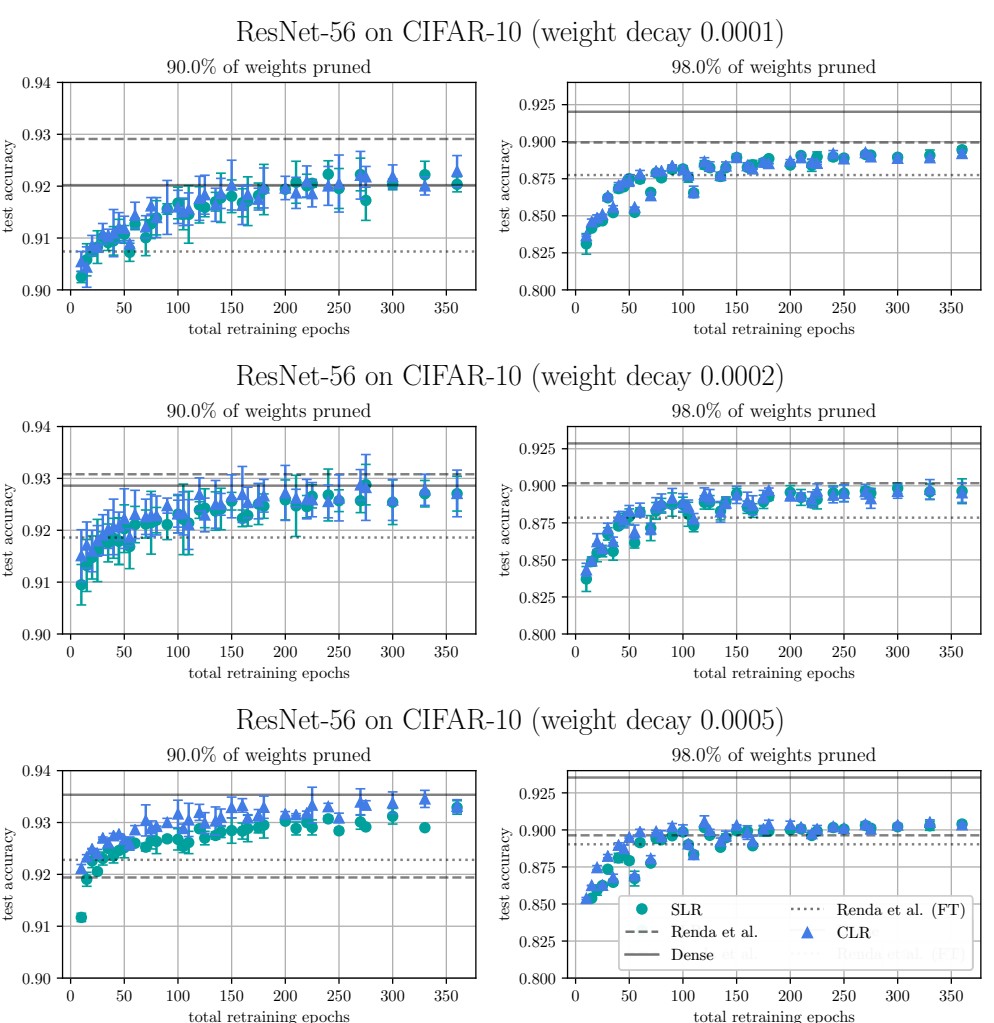

Figure 13: Test accuracy achieved by IMP in relation to the total number of epochs required for retraining using either CLR or SLR. Each row of the plots shows a different weight decay as indicated in the title. The pruned baselines were established by using the approach of Renda et al. (2020) using the respective weight decay and either in its original form, using LRW, or with a modified version using FT.

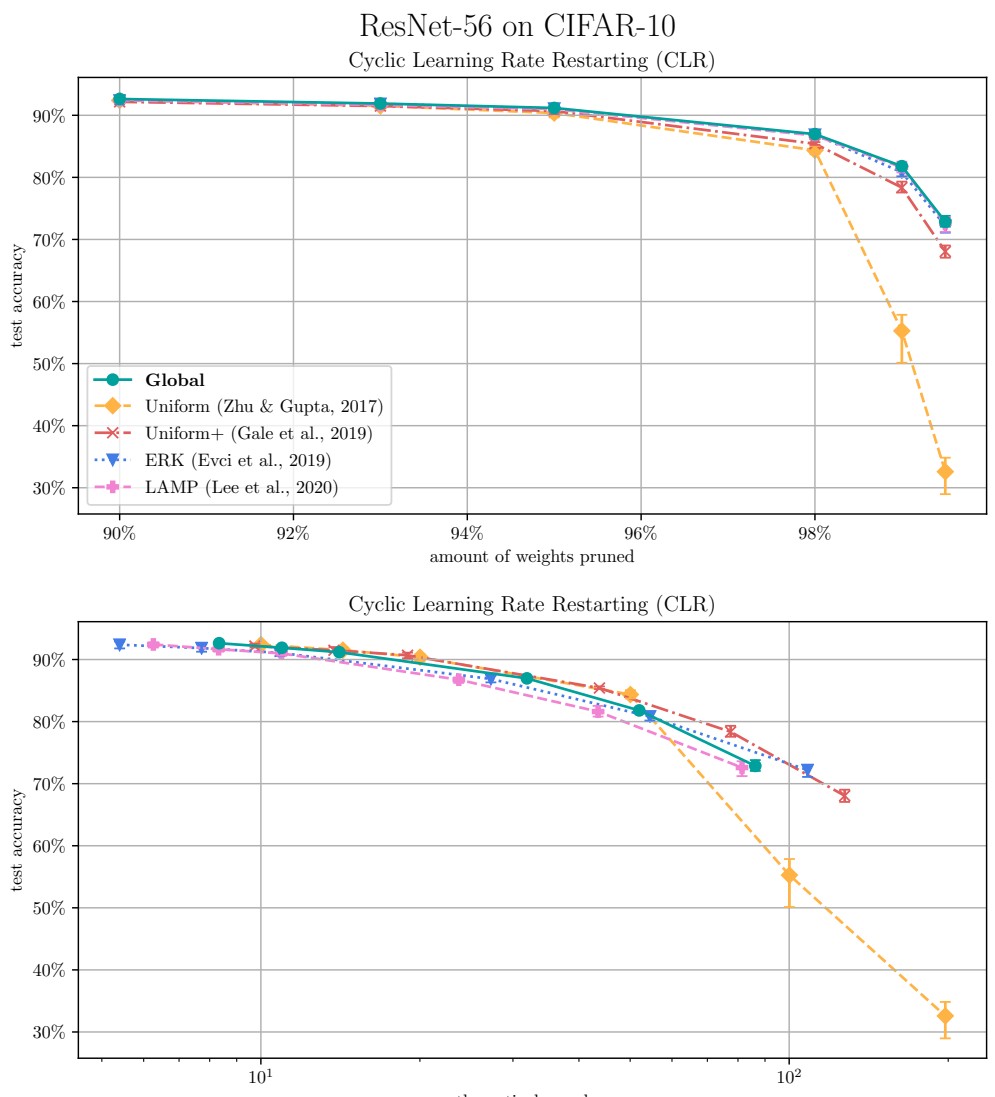

Figure 14: Performance-vs.-sparsity (above) and performance-vs.-theoretical-speedup (below) tradeoffs for ResNet-56 on CIFAR-10 for IMP in the One Shot setting with CLR as the retraining method. Each line corresponds to one pruning selection approach of interest. The model is retrained for 30 epochs after pruning.

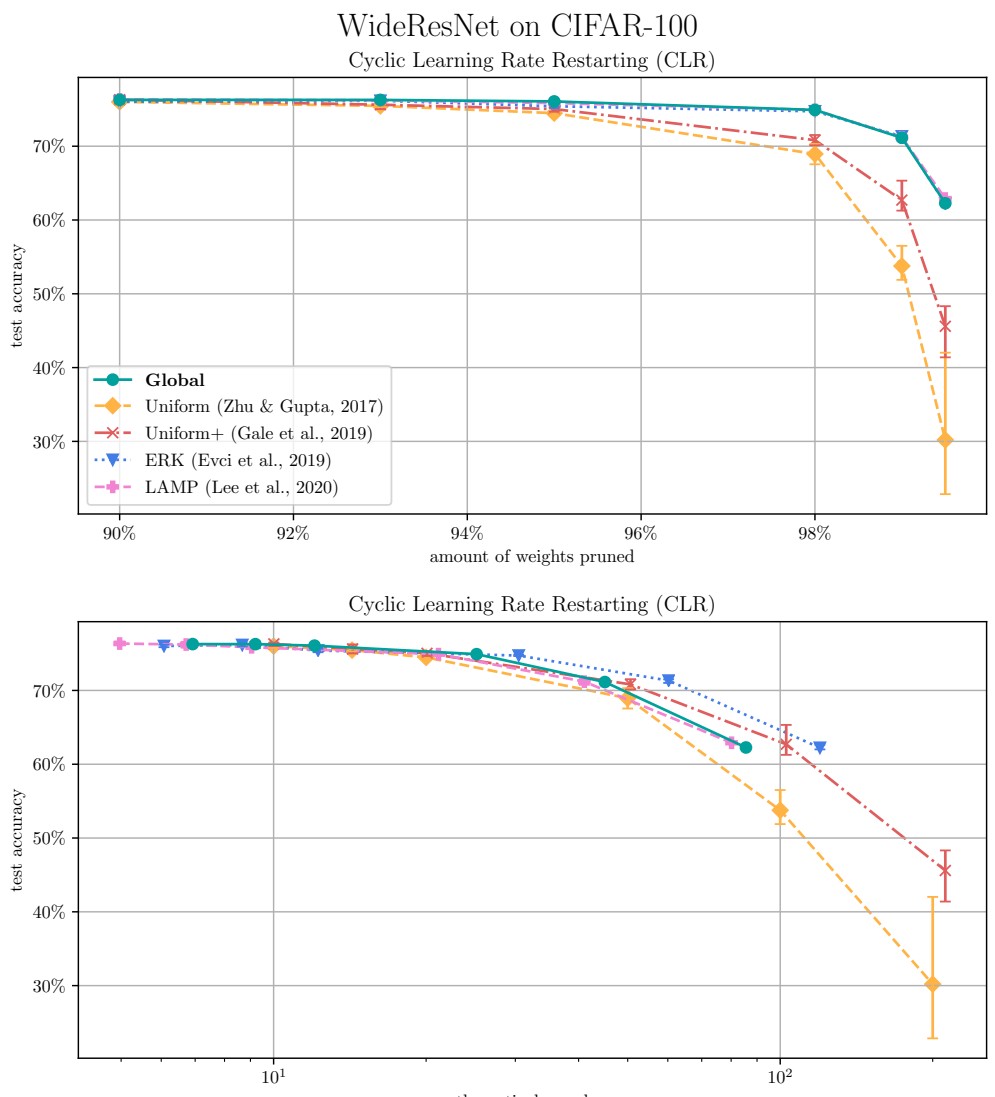

Figure 15: Performance-vs.-sparsity (above) and performance-vs.-theoretical-speedup (below) tradeoffs for WideResNet on CIFAR-100 for IMP in the One Shot setting with CLR as the retraining method.E ach line corresponds to one pruning selection approach of interest. The model is retrained for 30 epochs after pruning.

