# OpenReview forum: "Back to Basics: Efficient Network Compression via IMP"
_ICLR.cc/2022/Conference — ICLR 2022 Submitted_

### Official Review · Reviewer_WMwJ · 2021-11-02

**Correctness:** 3
**Technical Novelty And Significance:** 2
**Empirical Novelty And Significance:** 2
**Recommendation:** 5
**Confidence:** 4

**Main Review:**

Strengths
- The clarity in writing throughout most of the paper helps the reader quickly assimilate the contents and understand the intent of the paper.
- The level of detail, across the experiments as well as the figures, provided is of high quality and much appreciated.
- Explanation of the different approaches to pruning, their background and clearly advocating for the missing link is commendable.

Weaknesses
- The first paragraph on the arguments against IMP (Pg. 2, bullet point 1), reads slightly incoherently, like a mix of multiple ideas. A more clear and point by point description of sub-optimal states, re-training epochs and others would be extremely helpful.
- Additionally, "Many proposed improvements..." is followed by a single citation in the same bullet point 1. Adding multiple citations would be more appropriate with that statement.
- Pg. 4 Paragraph of Pruning Approaches: The nomenclature of "pruning approaches" is slightly counterintuitive when discussing ideas related to how pruning methods limit the overall compression of different layers. An alternative heading would summarize the contents more appropriately.
- While the intention of Section 3.1 is to highlight the low computational overhead of IMP in matching baseline performance, the intermediate outcome of SLR being better than FT traces the takeaways from Renda et. al (2020) and slightly weakens the contribution. Further, it isn't exactly clear what is the overhead when comparing the final set of results for IMP vs. SOTA pruning approaches. Comparison to a potentially weak baseline isn't quite enough and I encourage the authors to provide those results in a bid to strengthen their argument.
- Some of the key takeaways from Section 3.2 are very similar to that of Tanak et al. (2020) and Lee et al.(2020). Specifically, that of layer-collapse and the SLR vs.FT performance comparison. Leaning on these outcomes, in text and across image captions, weakens the novelty of the content. I encourage the authors to emphasize on IMP-specific behavior and insights.
- In Section 3.3, the last paragraph, the statement of IMP can be applied to already trained models while other methods in the experiment cannot isn't fair since the experiment uses pruning-during training approaches and not the general set of pruning methods.

After Rebuttal

I would like to thank the authors for their timely and pertinent discussions and revisions to the manuscript.
As I described previously the consistent references and experimental structure borrowed from existing work hinder the novelty of the work.
The authors have consistently clarified the difference in setup and mentioned that the exact outcomes differ from already existing work.
While the implementation and proposed work are focused on resurrecting IMP as a valid SOTA baseline, more in-depth work in alternative settings (unseen in existing literature) would be one possible way to help further highlight the novelty aspect of IMP.


**Summary Of The Paper:**

This work focuses on highlighting the strengths of Iterative Magnitude Pruning (IMP). Specifically, that it is capable of achieving strong performance when compared to more complex pruning approaches. The work explores the common arguments against IMP like, a) it reaches sub-optimal states since training doesn't compensate for sparse structures, b) it fails to identify optimal layer-wise pruning ratios and c) it is expensive, slow and non-competitive. The critical outcome shown is that IMP, with a global selection criterion and extremely small overhead, remains highly competitive with common state-of-the-art pruning approaches, both in sparsity, performance and theoretical speedup.

**Summary Of The Review:**

The proposed work does an extremely good job of explaining the landscape of pruning and setting up the missing link of IMP not being explored sufficiently. However, the experimental takeaways follow similar patterns to Tanaka et al.(2020) and Renda et al.(2020), which weaken the novelty of the work. The work sounds more like an extension of IMP into existing experimental frameworks and results than a purely novel instance.

---

> ### Author Response · Authors · 2021-11-11
> **Regarding the review WMwJ (1)**
>
> Thank you for your comments and thoughtful review. In the following we wish to address some of your individual points directly.
>
> > The first paragraph on the arguments against IMP (Pg. 2, bullet point 1), reads slightly incoherently, like a mix of multiple ideas. A more clear and point by point description of sub-optimal states, re-training epochs and others would be extremely helpful.
>
> Thank you for this comment, we do in fact agree that bullet point 1 needed to be rewritten to more concisely emphasise its main point and have done so in the latest revision, see the general comment. Stated succinctly, the consensus we are trying to summarise here is that there is some inherent benefit to having an implicit bias in the pruning method that goes beyond the direct consequence of achieving pruning stability. In fact, the main intent of this paper, as stated explicitly in the abstract and the introduction, is to question these claims through our experimental results in Section 3.3. We also note that some of that perceived vagueness or incoherence of this point is very much present in the type of claims made against IMP or in favour of pruning stability that are being summarised here.
>
> > Additionally, "Many proposed improvements..." is followed by a single citation in the same bullet point 1. Adding multiple citations would be more appropriate with that statement.
>
> This particular citation was referring to the preceding quote. Another example of this claim is the citation of Carreira-Perpiñán & Idelbayev. We felt it important not to blanket cite papers at this particular point, but to instead stick to direct attributed quotes.
>
> > Pg. 4 Paragraph of Pruning Approaches: The nomenclature of "pruning approaches" is slightly counterintuitive when discussing ideas related to how pruning methods limit the overall compression of different layers. An alternative heading would summarize the contents more appropriately.
>
> Thank you for this point, we have changed that title to “Pruning selection criteria” in the latest revision.
>
> > While the intention of Section 3.1 is to highlight the low computational overhead of IMP in matching baseline performance, the intermediate outcome of SLR being better than FT traces the takeaways from Renda et. al (2020) and slightly weakens the contribution. Further, it isn't exactly clear what is the overhead when comparing the final set of results for IMP vs. SOTA pruning approaches. Comparison to a potentially weak baseline isn't quite enough and I encourage the authors to provide those results in a bid to strengthen their argument.
>
> We think that there is some confusion at this point that stems from the fact that our submission did not sufficiently emphasise its main claim: the ultimate purpose of Section 3.1 but also of Section 3.2 is to establish how precisely IMP can develop its respective full potential without the thousands of retrain epochs that are commonly assumed to be essential. The outcome of these two sections are then applied in Section 3.3.
>
> The comparison between FT and SLR in Figure 1 does of course overlap with results stated by Renda et al. and Le & Hua, but the comparison to the pruned baseline also establishes that “IMP (…) seems to achieve most of its respective potential with significantly less than the total number of retraining epochs usually budgeted for its full iterative form.” This takeaway might seem obvious to some, but to our knowledge has not formally been established in the literature and is relevant to the ultimate conclusion of our paper. It in fact also seems to hold independent of the particular learning rate scheme during retraining, as can bee seen in our latest revision of the paper.
>
> This also addresses to what extend the pruned baseline used in Figure 1 is “weak”:  this plot is limited to IMP and its own potential, the comparison to other methods follows in Section 3.3. Note that we specifically address this by stating that “while it can be debated whether this approach (that is the pruned baseline by Renda et. al) reaches state-of- the-art results, it still serves as a good benchmark for the current potential of IMP.” We therefore also deemed this part relevant enough to include in the main text, rather than relegating it to the appendix.

---

> ### Author Response · Authors · 2021-11-11
> **Regarding the review WMwJ (2)**
>
> *This is a continuation of our previous comment.*
>
> > Some of the key takeaways from Section 3.2 are very similar to that of Tanaka et al. (2020) and Lee et al.(2020). Specifically, that of layer-collapse and the SLR vs.FT performance comparison. Leaning on these outcomes, in text and across image captions, weakens the novelty of the content. I encourage the authors to emphasize on IMP-specific behavior and insights.
>
> We refer to the previous comments regarding this point, but would also like to emphasise here that we were unable to reproduce the results of Lee et al. in Figure 2, as we explicitly discuss. This is why we advocate for the global selection criterion and include this figure in the main text to motivate this. Had the results been more in line with previously published results, those computational results would have most likely ended up in the appendix.
>
> Regarding the phenomenon of layer collapse, we agree however that the result presented in Figure 3 somewhat overlaps with that of Figure 5 (a) by Tanaka et al. and thank you for pointing this out. It should be noted though, that Tanaka et al. studied this phenomenon when pruning _before_ training whereas we observe similar results for pruning _after_ training. We have still moved it to the appendix in our latest revision and used the additional space to build out some of the other aspects raised by you.
>
> > In Section 3.3, the last paragraph, the statement of IMP can be applied to already trained models while other methods in the experiment cannot isn't fair since the experiment uses pruning-during training approaches and not the general set of pruning methods.
>
> Could you please elaborate on this point? The intended purpose of our submission was to compare pruning stable methods with respect to the very aspects in which they are commonly claimed to have the advantage over such a simple pruning unstable method as IMP, namely the achieved sparsity-performance-tradeoff and the computational costs required. The fact that IMP (here as a generic stand in for pruning unstable methods)  has the mentioned additional benefits on top of performing on par with SOTA pruning stable methods “on their home turf” seemed relevant to include at this point.
>
> > The proposed work does an extremely good job of explaining the landscape of pruning and setting up the missing link of IMP not being explored sufficiently. However, the experimental takeaways follow similar patterns to Tanaka et al.(2020) and Renda et al.(2020), which weaken the novelty of the work. The work sounds more like an extension of IMP into existing experimental frameworks and results than a purely novel instance.
>
> We thank you for seeing the positive aspects of our submission, but hope that we could also at least somewhat convince you of the relevance of what we intended to show with our submission. We explicitly acknowledge in our paper that we are building on the works of Renda et al., Le & Hua, and also Tanaka et al. by “putting an additional spotlight on the total computational cost of IMP in a direct comparison with methods that are commonly assumed to outperform IMP in that aspect by avoiding retraining.” To not suggest yet another learning rate scheme, pruning selection criterion or some other modification of IMP was very much intentional on our part for this particular comparative study. If you have any further suggestions as to how we can more precisely argue or demonstrate our main point (or if you believe that the argument can be weakened through specific further experiments) we would be very glad to hear them.

---

> > ### Comment · Reviewer_WMwJ · 2021-11-18
> > **Response to extended comments**
> >
> > I appreciate the authors' detailed response to the feedback provided.
> > - The comment related to the statement in Section 3.3 was meant to highlight the fact that pruning-during-training approaches are handicapped in training from scratch while IMP is flexible. The intent was to ensure that the statement was only distinguishing aspects of the work being compared against.
> > - As a general comment, the proposed work is extremely thorough about the key aspects being addressed and doesn't leave much of a "gray area" in terms of the content. However, when discussing the results there is a strong comparative element, wrt. existing work. The nature of the topic and ideas at hand force this kind of style yet this is precisely what weakens the novelty of the contributions. I would encourage your to experiment with re-arranging the discussion (Highlight the results themselves (the novelty), compare against existing work and how it matches/bucks the expected trends).
> > - For the results in Section 3.3, originally the takeaways from Section 3.1 and 3.2 highlighted the claims made at the beginning of the manuscript but Section 3.3 did not possess the metric of epochs to gauge the reduction in time/iterations for the best version of IMP. The updated Table 1 goes some way to highlighting this but a concrete discussion around that point (and visual cues in the table possibly) would further highlight it. I believe it is critical since this is one of the key contributions stated.
> > - Finally, in Table 1 I would encourage the authors to indicate the expected increase and decrease in values attributed to good methods for each metric.

---

> > > ### Author Response · Authors · 2021-11-19
> > > **Changes in the latest revision**
> > >
> > > Thank you for your response. We have just uploaded another revision that, along with some additional computational results, hopefully addresses your concerns. In particular, we have updated the abstract and partially restructured and rewritten the introduction to more strongly emphasize the main intent and novel findings of our work. We have also removed the paragraph highlighting the flexibility of IMP from Section 3.3. We hope this new structure more clearly shows that the findings of our work are in fact novel and directly refute some commonly held beliefs in the literature.
> > >
> > > Regarding your points about Table 1, could you please also clarify how you think the presentation can be improved here? In particular, we are uncertain what you are referring to regarding the "metric of epochs" as well as the "expected increase and decrease in values attributed to good methods for each metric"?

---

### Official Review · Reviewer_fSGB · 2021-11-03

**Correctness:** 3
**Technical Novelty And Significance:** 2
**Empirical Novelty And Significance:** 4
**Recommendation:** 6
**Confidence:** 3

**Main Review:**

This paper is motivated by a simple question: whether the basic IMP method is enough for weights pruning? Through the whole paper, the authors conducted lots of experiments to show that IMP with a good retraining is as good as many recently proposed pruning methods. The authors also show that using IMP we can also directly determine the layerwise sparsity through global ranking, instead of using more complex method to determine the sparsity for each layer.

Although the authors did lots of experiments validating the claim, the experiments are kind of limited. All the experiments are in image classification, with larger size models. I am wondering how is the result on compact models like mobilenet. Besides CNNs, weights pruning are actually used for other architectures too, e.g., LSTMs and transformers. How are the comparison on those cases? I admit that this paper already conducted a comprehensive comparison, while it's hard to say IMP is enough for pruning with only a few models on Image classifications showing that. In addition, many new pruning algorithms are actually built for NLP and speech tasks.

**Summary Of The Paper:**

This paper investigated several recently proposed pruning stable approaches and compared them with the basic iterative magnitude based pruning (IMP), and observe that IMP actually performs on-par or even better in the experiments. The authors investigated the retraining approaches, the computation cost, the importance of sparsity allocation (including layer-collapse) and compared IMP (+retraining) to different pruning methods.

**Summary Of The Review:**

I like the author's view and the motivation to make a realizable and unified baseline for weights pruning. However, I think it's better to have more evidence on different models and tasks to validate this conclusion. If the proposed claim can be further validated, I think it will be a significant contribution to the community.

---

> ### Author Response · Authors · 2021-11-11
> **Regarding the review fSGB**
>
> Thank you for your comments and thoughtful review. We fully agree, and already acknowledge in the Discussion section of our paper, that the claims of our paper are of course limited to image classification tasks and common convolutional neural network architectures. Extending these results, in particular to NLP tasks and appropriate network architectures, is extremely interesting. However, we would like to note that many of the prior works that we are explicitly referring to and building upon, likewise restricted their focus to image classification tasks. In particular, Blalock et al., to whom we refer regarding best practices for a fair computational evaluation, note that “image classification is only one of the countless tasks to which neural networks have been applied. However, because the overwhelming majority of papers in our corpus focus on these metrics, our meta-analysis necessarily does as well.” Many of the methods included in our comparative study use the same general setup when arguing their respective efficacy, while commonly including significantly fewer methods for comparison than are contained in our results.
>
> Regardless, we are currently finishing up additional CIFAR-100 results for Section 3.3 and intend to have them ready and available as a revision before the end of the rebuttal period. Likewise, we are hopeful to be able to include NLP based experimental results for a potential camera ready version of this submission, though this certainly exceeds what we can include during this rebuttal period.
>
> Regarding the inclusion of the MobileNet architecture, we are definitely also interested in exploring other architectures for image classification tasks. We note though that with ResNet-56, ResNet-50, WRN28x10 and VGG-16 our results already span architectures from as little as 850K to as much as 138 Mio parameters. Adding the MobileNet architecture with its 13 Mio parameters to either the CIFAR-100 or ImageNet results still seems like a relevant addition in the future.

---

> ### Author Response · Authors · 2021-11-19
> **Additional results regarding CIFAR-100**
>
> Thank you again for your comment. As already mentioned in our previous comment, we would like to point out that we have added additional computational results in the latest revision of our submission.

---

### Official Review · Reviewer_dL1d · 2021-11-03

**Correctness:** 3
**Technical Novelty And Significance:** 2
**Empirical Novelty And Significance:** 3
**Recommendation:** 5
**Confidence:** 4

**Main Review:**


Strength:

1. The paper revisits IMP, a basic yet important pruning approach. The empirical findings are promising: IMP can be trained on par with stable pruning approaches using reasonable training time under proper learning rate schemes.

2. Extensive studies with detailed experimental setups are provided. Code is provided for reproducibility.


Weakness:

1. It is not explained why SLR is combined. I wonder if IMP can be improved when combined with other training tricks (e.g., CLR) w.r.t. the discussed aspects. And will there be any improvement if other baselines (e.g., Uniform/ERK/LAMP in Figure 2) are combined with SLR?

2. The writing is not very focused, making it kind of hard to follow by the general audience, especially for Section 2. More technical details can be introduced for IMP and SLR, both of which are studied throughout the experiments.

3. While the paper provides empirical studies for IMP, the technical contribution can be minor, as most components are existing methodologies.


Detailed comments:

- The title can be misleading? The empirical results show that only under proper learning rate schemes (SLR) can IMP achieve better performance with less training time. However, SLR is not a basic technique.

- While the authors draw the conclusions based on the improved accuracy with SLR, the reasons for improvement are still unclear nor well explained. It would be more inspiring to explicitly study why previous IMP are sub-optimal, and how SLR can improve IMP.

- Are the rest baselines in Figure 1-3 trained with the same learning rate scheme (FT or SLR)? Otherwise, it can be unfair to compare with IMP+SLR.

- Reference formats are not formal and inconsistent.

- It can be restrictive to study unstructured pruning as well as theoretical speed-up with FLOPs reduction in practice.

**Summary Of The Paper:**

The paper studies a fundamental and important research approach in network pruning: iterative magnitude pruning (IMP). Previously IMP is criticized to be time-consuming, layer-independent and sub-optimal in performance. In this paper, extensive empirical studies are conducted to show that under proper learning rates, IMP can have close performance with more advanced pruning approaches, with little training time increased.

**Summary Of The Review:**

The empirical findings are promising. However, it is still not clear in what way SLR can improve IMP, and the reasons behind the improved accuracy / reduced training time.


=== Post rebuttal ===
Thanks for the authors' response and paper revision, which addresses some of my concerns. While the paper introduces inspiring findings on how SLR (or CLR) help IMP, most components are from existing techniques. I choose to keep my score, and encourage the authors to provide more in-depth analysis behind the improvement by SLR, via either new methodologies or perspectives.

---

> ### Author Response · Authors · 2021-11-11
> **Regarding the review dL1d (1)**
>
> Thank you for your comments and thoughtful review. In the following we wish to address some of your individual points directly.
>
> > While the paper provides empirical studies for IMP, the technical contribution can be minor, as most components are existing methodologies.  (...) It would be more inspiring to explicitly study why previous IMP are sub-optimal, and how SLR can improve IMP.
>
> Several attempts have been made in the literature, in particular by Renda et al. and Le & Hua, to find explanations for the improved performance coming from specific learning rate schemes. Motivated by your point, we have likewise decided to add a discussion of that aspect to a current revision of our submission, see our general comment. We think, that the nature of the proposed retraining methods indicates that the retraining phase is, at its core, similar to the usual training phase. Following this rationale, the success of LRW, SLR and CLR over FT should be attributed to the fact that they stick to current best practices regarding the development of a successful learning rate scheme. We go into more detail of why a large initial and exponentially decaying learning rate has become the standard practice for regular training in our latest revision. Put more succinctly: LRW, SLR and CLR provide some good heuristic guidance for how to effectively do so without being swamped by an insurmountable amount of  decisions regarding hyperparameters. This is not intended to denigrate the contributions of Renda et al. and Le & Hua.; in fact, as we explicitly write, our paper is intended to compliment their results by extending the lessons that can be drawn from them.
>
> However, we emphasise that our intention behind this paper was not to derive a theoretical justification for one or the other method, or to even suggest a new one, but to instead establish that IMP with an existing learning rate scheme that is easily implementable and requires no additional parameterisation can outperform significantly more complex methods with little to no computational overhead.
>
> > The title can be misleading? The empirical results show that only under proper learning rate schemes (SLR) can IMP achieve better performance with less training time. However, SLR is not a basic technique.
>
> Tying in with our previous point, we believe that a modified learning rate scheme, the success of which we acknowledge and underline with our own results, does not change core aspects of the functionality of IMP as a pruning method.
>
> > It is not explained why SLR is combined. I wonder if IMP can be improved when combined with other training tricks (e.g., CLR) w.r.t. the discussed aspects. (...)
>
> We briefly address this point at the beginning of Section 3, where we refer to the fact that we intentionally limit ourselves to stepped learning rates throughout. To address your concerns, we have also added results using CLR for Section 3.1 and 3.2 to the appendix of the current revision along with a brief discussion that mirrors the arguments here. We do not find them to drastically impact the results either way.
>
> Note that not employing a drastically different type of learning rate scheme for the retrain phase of IMP in the form of CLR but instead sticking to SLR was important to us to ensure a fair comparison in Section 3.3. It is unclear to us if the positive impact of cosine annealing in CLR  has anything to do with the fact that it is being used in the retraining phase of a pruning method, or if the original dense training or even the training of a pruning stable method wouldn't likewise benefit from it. Using CLR in the retraining of IMP but sticking to stepped learning rates for the initial training as well as the pruning stable methods, would make the results of Section 3.3 not an apples-to-apples comparison.
>
> > And will there be any improvement if other baselines (e.g., Uniform_ERK_LAMP in Figure 2) are combined with SLR?
>
> This is very directly and explicitly explored in Section 3.1 where each of the selection schemes (including the global one) is both combined with FT and SLR (respectively the left- and right-hand-side of Figure 2). The motivation for doing so is also addressed in Section 2.
>
> > Are the rest baselines in Figure 1-3 trained with the same learning rate scheme (FT or SLR)? (...)
>
> Regarding Figure 1: the dense baseline is trained using the stepped learning rate scheme as indicated in Table 1 and the pruned baseline is due to Renda, as stated in Section 3.1 as well as the caption of Figure 1, and therefore, as stated in Section 2, uses LRW for retraining. Note that in this context LRW = SLR as observed in Section 2. Regarding Figures 2 and 3, could you please clarify what you are referring to as a baseline in this context?

---

> ### Author Response · Authors · 2021-11-11
> **Regarding the review dL1d (2)**
>
> *This is a continuation of our previous comment.*
>
> > The writing is not very focused, making it kind of hard to follow by the general audience, especially for Section 2. More technical details can be introduced for IMP and SLR, both of which are studied throughout the experiments.
>
> We appreciate that feedback. Could you please give us some specific recommendations on how we could improve the clarity of our writing, in particular in Section 2? The purpose of this section is to set up the necessary background for the computational results in Section 3. Given the amount of methods compared, we necessarily had to cut some exposition short. However, in particular the introduction of IMP and SLR seems about on par in length with those in the respective papers where they are proposed. They are intentionally simple methods after all and hence do not require a great deal of motivation.
>
> > Reference formats are not formal and inconsistent.
>
> Thank you for pointing this out, we have since thoroughly checked and updated all references again.
>
> > It can be restrictive to study unstructured pruning as well as theoretical speed-up with FLOPs reduction in practice.
>
> We agree with your point that the focus on unstructured pruning somewhat limits our results and have added an acknowledgment of that fact to the Discussion section. However, many of the prior works that we are explicitly referring to and building upon, likewise restricted their focus to unstructured pruning. Could you also please clarify what you are referring to with the second part of your sentence?

---

### Author Response · Authors · 2021-11-11
**Summary of the rebuttal revision**

We would like to thank all of the reviewers for their insightful and helpful comments. Since submission, we have worked on further improving both results and the representation of our paper and have made additional changes in order to address particular points raised in this review process. In the following, we provide a list with all major or relevant changes:

**1. Introduction**
* As suggested by reviewer WMwj, we changed the paragraph containing the first common claim against IMP to more concisely emphasise its main point.
*  We have also slightly adjusted the paragraph listing the papers contributions.


**2. Overview of Pruning Methods and Methodology**
* We restructured the three main paragraphs into proper subsections. In particular, we renamed subsection 2.2 to "Pruning selection criteria" as suggested by Reviewer WMwj.
*   Subsection "Retraining approaches": We added a paragraph addressing the success of recently proposed retraining methods.

**3. Experimental results**
* We moved Table 1 containing the exact training setting of all architecture-dataset pairs to the appropriate section in the appendix.

*3.1 The Computational Cost of IMP*
* We extended Figure 1 by computing the pruned and dense baselines for three different weight decays from scratch to rule out any dependency on the weight decay factor. Additionally, we added an FT-based variant of the Renda et al. baseline. The plot now depicts the highest mean accuracy achieved by any parameter configuration, including weight decay. A breakdown into the different weight decay factors has been added to the appendix. Further, we added a comparison with CLR to the appendix as suggested by Reviewer dL1d. We also slightly adjusted our interpretation of the results accordingly.

*3.2 The Importance of the Sparsity Distribution*
* We moved Figure 3 to the appendix to address the concerns of Reviewer WMwJ. We highlight how our work differs from that of Tanaka et al. and also address the sparsity-vs.-speedup tradeoffs of the different methods in more detail.

*3.3 Comparing IMP to Pruning Stable Approaches*
* In order to better visualize our results, we moved an enhaced version of Figure 4 to the appendix and replaced it in the main part by a table listing Accuracy, Speedup and Sparsity for all methods at stake for three selected goal sparsities. The table also includes the relative times needed when comparing to training a dense model (previously we compared to the runtime of IMP). The full table can be found in the appendix.

---

### Author Response · Authors · 2021-11-19
**Summary of the second rebuttal revision**

As promised in the first rebuttal revision, we further improved the paper. In the following, we provide a list with all major or relevant changes:

**0. Abstract**
* We reformulated the abstract to highlight the novelty of our work.

**1. Introduction**
*  We slightly restructured the introduction to emphasize the motivation behind our work.

**3. Experimental results**

*3.3 Comparing IMP to Pruning Stable Approaches*
* To further underline our findings, we added a full comparison of IMP vs. pruning-during-training methods in the case of WideResNet on CIFAR-100. The corresponding table as well as plots can be found in the appendix. Further, we strengthened our CIFAR-10 results by additionally tuning the learning rate schedule and hence broadening the grid search for all methods.
* We rewrote the last paragraphs of the section, including the final discussion, to emphasize the strengths of IMP w.r.t. its computational efficacy and to highlight why we think that our work is novel and a significant contribution to the field, showing that the narrative of needing to impose strong implicit biases on the training routine is questionable.

---

### Decision · Program_Chairs · 2022-01-20

**Decision:**

Reject

**Comment:**

The paper provides an analysis of the well known method of Iterative Magnitude Pruning (IMP) for DNN compression. The problem tackled is undoubtably an important one, and IMP is likely one of the most known solutions for DNN compression. As such, there is no doubt that the paper is well motivated. In addition to the motivated task, the reviews indicate that the paper is well written and provides a thorough review of the related literature, making the paper easy to read and follow.
The main weakness of the paper seems to its novelty, as it seems that similar analyses have been done in the past. This issue was raised by the reviews and remained after the correspondence with the authors:

 WMeJ: “As I described previously the consistent references and experimental structure borrowed from existing work hinder the novelty of the work”, dL1d: “While the paper introduces inspiring findings on how SLR (or CLR) help IMP, most components are from existing techniques”.

Given the discussion and concerns related to the novelty of the paper, I feel that the paper requires too major of a revision to be accepted, either improving its core analysis, or presenting it in a better way that clearly distinguishes it from previous art.